**eLife** RESEARCH ARTICLE

# Auditory cortex learns to discriminate audiovisual cues through selective multisensory enhancement

**Song Chang[1], Beilin Zheng[2], Les Keniston[3], Jinghong Xu[1]\*, Liping Yu[1]\***

[1]Key Laboratory of Brain Functional Genomics (Ministry of Education and Shanghai), School of Life Sciences, East China Normal University, Shanghai, China; [2]College of Information Engineering, Hangzhou Vocational and Technical College, Hangzhou, China; [3]Department of Biomedical Sciences, Kentucky College of Osteopathic Medicine, University of Pikeville, Pikeville, United States

## eLife Assessment

This is an **important** study that aims to investigate the behavioral relevance of multisensory responses recorded in the auditory cortex. The experiments are elegant and well-designed and are supported by appropriate analyses of the data. Although **solid** evidence is presented that is consistent with learning-dependent encoding of visual information in auditory cortex, further work is needed to establish the origin and nature of these non-auditory signals and to definitively rule out any effects of movement-related activity.

**\*For correspondence:**
jhxu@bio.ecnu.edu.cn (JX);
lpyu@bio.ecnu.edu.cn (LY)

**Competing interest:** The authors declare that no competing interests exist.

**Abstract** Multisensory object discrimination is essential in everyday life, yet the neural mechanisms underlying this process remain unclear. In this study, we trained rats to perform a two-alternative forced-choice task using both auditory and visual cues. Our findings reveal that multisensory perceptual learning actively engages auditory cortex (AC) neurons in both visual and audiovisual processing. Importantly, many audiovisual neurons in the AC exhibited experience-dependent associations between their visual and auditory preferences, displaying a unique integration model. This model employed selective multisensory enhancement for the auditory-visual pairing guiding the contralateral choice, which correlated with improved multisensory discrimination. Furthermore, AC neurons effectively distinguished whether a preferred auditory stimulus was paired with its associated visual stimulus using this distinct integrative mechanism. Our results highlight the capability of sensory cortices to develop sophisticated integrative strategies, adapting to task demands to enhance multisensory discrimination abilities.

## Introduction

In our daily lives, the integration of visual and auditory information is crucial for detecting, discriminating, and identifying multisensory objects. When faced with stimuli like seeing an apple while hearing the sound 'apple' or 'peach' simultaneously, our brain must synthesize these inputs to form perceptions and make judgments based on the knowledge of whether the visual information matches the auditory input. To date, it remains unclear exactly where and how the brain integrates across-sensory inputs to benefit cognition and behavior. Traditionally, higher association areas in the temporal, frontal, and parietal lobes were thought to be pivotal for merging visual and auditory signals. However, recent research suggests that even primary sensory cortices, such as auditory and visual cortices, contribute

significantly to this integration process (*Perrodin et al., 2015*; *Ghazanfar et al., 2005*; *Kayser et al., 2008*; *Bizley et al., 2007*; *Atilgan et al., 2018*; *Meijer et al., 2017*; *Ibrahim et al., 2016*).

Studies have shown that visual stimuli can modulate auditory responses in the AC via pathways involving the lateral posterior nucleus of the thalamus (*Chou et al., 2020*), and the deep layers of the AC serve as crucial hubs for integrating cross-modal contextual information (*Morrill et al., 2022*). Even irrelevant visual cues can affect sound discrimination in AC (*Chang et al., 2022*). Anatomical investigations reveal reciprocal nerve projections between auditory and visual cortices (*Bizley et al., 2007*; *Stehberg et al., 2014*; *Falchier et al., 2002*; *Cappe and Barone, 2005*; *Campi et al., 2010*; *Banks et al., 2011*), highlighting the interconnected nature of these sensory systems. Moreover, two-photon calcium imaging in awake mice has shown that audiovisual encoding in the primary visual cortex depends on the temporal congruency of stimuli, with temporally congruent audiovisual stimuli eliciting balanced enhancement and suppression, whereas incongruent stimuli predominantly result in suppression (*Meijer et al., 2017*). However, despite these findings, the precise mechanisms by which sensory cortices integrate cross-sensory inputs for multisensory object discrimination remain unknown.

Previous research on cross-modal modulation has predominantly focused on anesthetized or passive animal models (*Kayser et al., 2008*; *Atilgan et al., 2018*; *Meijer et al., 2017*; *Stein et al., 2014*; *Deneux et al., 2019*), exploring the influence of stimulus properties and spatiotemporal arrangements on sensory interactions (*Kayser et al., 2008*; *Meijer et al., 2017*; *Deneux et al., 2019*; *Wallace et al., 2020*; *Xu et al., 2018*). However, sensory representations, including multisensory processing, are known to be context-dependent (*Elgueda et al., 2019*; *Han et al., 2022*; *Otazu et al., 2009*) and can be shaped by perceptual learning (*Xu et al., 2014*; *Vincis and Fontanini, 2016*; *Knöpfel et al., 2019*). Therefore, the way sensory cortices integrate information during active tasks may differ significantly from what has been observed in passive or anesthetized states. Relatively few studies have investigated cross-modal interactions during the performance of multisensory tasks, regardless of the brain region studied (*Garner and Keller, 2022*; *Raposo et al., 2014*; *Hirokawa et al., 2011*). This limits our understanding of multisensory integration in sensory cortices, particularly regarding: (1) Do neurons in sensory cortices adopt consistent integration strategies across different audiovisual pairings, or do these strategies vary depending on the pairing? (2) How does multisensory perceptual learning reshape cortical representations of audiovisual objects? (3) How does the congruence between auditory and visual features—whether they 'match' or 'mismatch' based on learned associations—impact neural integration?

We investigated this by training rats on a multisensory discrimination task involving both auditory and visual stimuli. We then examined cue selectivity and auditory-visual integration in AC neurons of these well-trained rats. Our findings demonstrate that multisensory discrimination training fosters experience-dependent associations between auditory and visual features within AC neurons. During task performance, AC neurons often exhibited multisensory enhancement for the preferred auditory-visual pairing, with no such enhancement observed for the non-preferred pairing. Importantly, this selective enhancement correlated with the animals' ability to discriminate the audiovisual pairings. Furthermore, the degree of auditory-visual integration was linked to the congruence of auditory and visual features. Our findings suggest AC plays a more significant role in multisensory integration than previously thought.

## Results

### Multisensory discrimination task in freely moving rats

To investigate how AC neurons integrate visual information into audiovisual processing, we trained 10 adult male Long Evans rats on a multisensory discrimination task (*Figure 1a*). During the task, the rat initiated a trial by inserting its nose into the central port, which triggered a randomly selected target stimulus from a pool of six cues: two auditory (3 kHz pure tone, $A_{3k}$; 10 kHz pure tone, $A_{10k}$), two visual (horizontal light bar, $V_{hz}$; vertical light bar, $V_{vt}$), and two multisensory cues ($A_{3k}V_{hz}$, $A_{10k}V_{vt}$). Based on the cue, the rats had to choose the correct left or right port for a water reward within 3 s after the stimulus onset. Incorrect choices or no response resulted in a 5-s timeout. The training proceeded in two stages. In the first stage, which typically lasted 3–5 weeks, rats were trained to discriminate between two audiovisual cues. In the second stage, an additional four unisensory cues were introduced, training

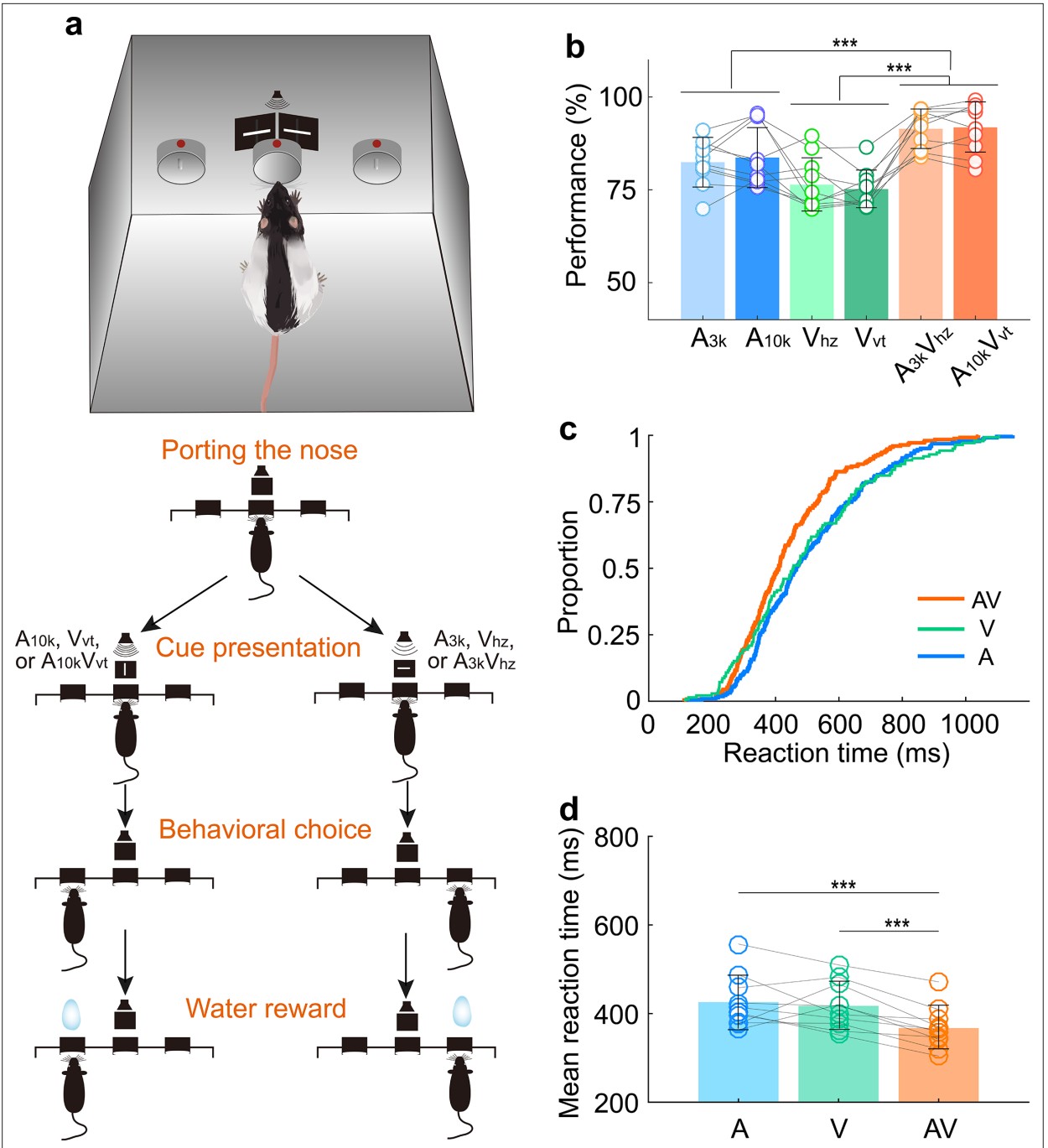

**Figure 1.** Multisensory discrimination task for rats. (**a**) Schematic illustration of the behavioral task. A randomly selected stimulus was triggered when a rat placed its nose in the central port. If the triggered stimulus was a 10 kHz pure tone ($A_{10k}$), a vertical light bar ($V_{vt}$), or their combination ($A_{10k}V_{vt}$), the rat was rewarded at the left port. Conversely, for other stimuli (a 3 kHz pure tone ($A_{3k}$), a horizontal light bar ($V_{hz}$), or $A_{3k}V_{hz}$), the rat was required to move to the right port. Visual stimuli were presented via custom-made LED matrix panels, with one panel for each side. Auditory stimuli were delivered through a centrally located speaker. (**b**) The mean performance for each stimulus condition across rats. Circles connected by a gray line represent data from one individual rat. (**c**) Cumulative frequency distribution of reaction time (time from cue onset to leaving the central port) for one representative rat in auditory, visual and multisensory trials (correct only). (**d**) Comparison of average reaction times across rats in auditory, visual, and multisensory trials (correct only). ***, $p < 0.001$. Error bars represent SDs.

the rats to discriminate a total of six cues. To ensure reliable performance, rats needed to achieve 80% accuracy (including at least 70% correct in each modality) for three consecutive sessions (typically taking 2–4 months to achieve this level of accuracy).

Consistent with previous studies (*Chang et al., 2022*; *Raposo et al., 2014*; *Angelaki et al., 2009*; *Smyre et al., 2021*), rats performed better when responding to combined auditory and visual cues (multisensory trials) compared to trials with only auditory or visual cue (*Figure 1b*). This suggests that multisensory cues facilitate decision-making. Reaction time, measured as the time from cue onset to when the rat left the central port, was shorter in multisensory trials (*Figure 1c and d*). This indicates that multisensory processing in the brain helps rats discriminate between cues more efficiently (mean reaction time across rats, multisensory, 367±53ms; auditory, 426±60ms; visual, 417±53ms; multisensory vs. each unisensory, p<0.001, paired t-test. *Figure 1d*).

## Auditory, visual, and multisensory discrimination of AC neurons in multisensory discrimination task

To investigate the discriminative and integrative properties of AC neurons, we implanted tetrodes in the right primary auditory cortex of well-trained rats (n=10; *Figure 2a*). Careful implantation procedures were designed and followed to minimize neuron sampling biases. The characteristic frequencies of recorded AC neurons, measured immediately after tetrode implantation, spanned a broad frequency range (*Figure 2—figure supplement 1*). We examined a total of 559 AC neurons (56±29 neurons per rat) that responded to at least one target stimulus during task engagement. Interestingly, a substantial proportion of neurons (35%, 196/559) showed visual responses (*Figure 2b*), which was notably higher than the 14% (14%, 39/275, $\chi^2$ = 27.5, p < 0.001) recorded in another group of rats (n=8) engaged in a free-choice task where rats were not required to discriminate triggered cues and could receive water rewards with any behavioral response (*Figure 2b*). This suggests multisensory discrimination training enhances visual representation in the auditory cortex. To optimize the alignment of auditory and visual responses and reveal the greatest potential for multisensory integration, we used long-ramp pure tone auditory stimuli and quick LED-array-elicited visual stimuli. While auditory responses were still slightly earlier than visual responses (*Figure 2—figure supplement 2*), the temporal alignment was sufficient to support robust integration. Notably, 27% (150/559) of neurons responded to both auditory and visual stimuli (audiovisual neurons), and a small number (n=7) only responded to the auditory-visual combination (*Figure 2b*).

During task engagement, AC neurons displayed a clear preference for one target sound over the other. As exemplified in *Figure 2c and d*, many neurons exhibited a robust response to one target sound while showing a weak or negligible response to the other. We quantified this preference using receiver operating characteristic (ROC) analysis. Since most neurons exhibited their main cue-evoked response within the initial period of cue presentation (*Figure 2e*), our analysis focused on responses within the 0–150ms window after cue onset. We found a significant majority (61%, 307/506) of auditory-responsive neurons exhibited this selectivity during the task (*Figure 2f*). Notably, most neurons favored the high-frequency tone ($A_{10k}$ preferred: 261; $A_{3k}$ preferred: 46, *Figure 2f*). This aligns with findings that neurons in the AC and medial prefrontal cortex selectively preferred the tone associated with the behavioral choice contralateral to the recorded cortices during sound discrimination tasks (*Chang et al., 2022*; *Zheng et al., 2021*), potentially reflecting the formation of sound-to-action associations (*Xiong et al., 2015*). However, this preference represents a neural correlate, and further work is required to establish its causal link to behavioral choices. Such pronounced sound preference and bias were absent in the free-choice group (*Figure 2e and f*), suggesting it is directly linked to active discrimination. Anesthesia decreased auditory preference (*Figure 2—figure supplement 3*), further supporting its dependence on active engagement.

Regarding the visual modality, 41% (80/196) of visually-responsive neurons showed a significant visual preference (*Figure 2f*). The visual responses observed within the 0–150ms window after cue onset were consistent and unlikely to result from visually evoked movement-related activity. This conclusion is supported by the early timing of the response (*Figure 2e*) and exemplified by a neuron with a low spontaneous firing rate and a robust, stimulus-evoked response (*Figure 2—figure supplement 4*). Similar to the auditory selectivity observed, a greater proportion of neurons favored the visual stimulus ($V_{vt}$) associated with the contralateral choice, with a 3:1 ratio of $V_{vt}$-preferred to $V_{hz}$-preferred neurons. This convergence of auditory and visual selectivity likely results from multisensory

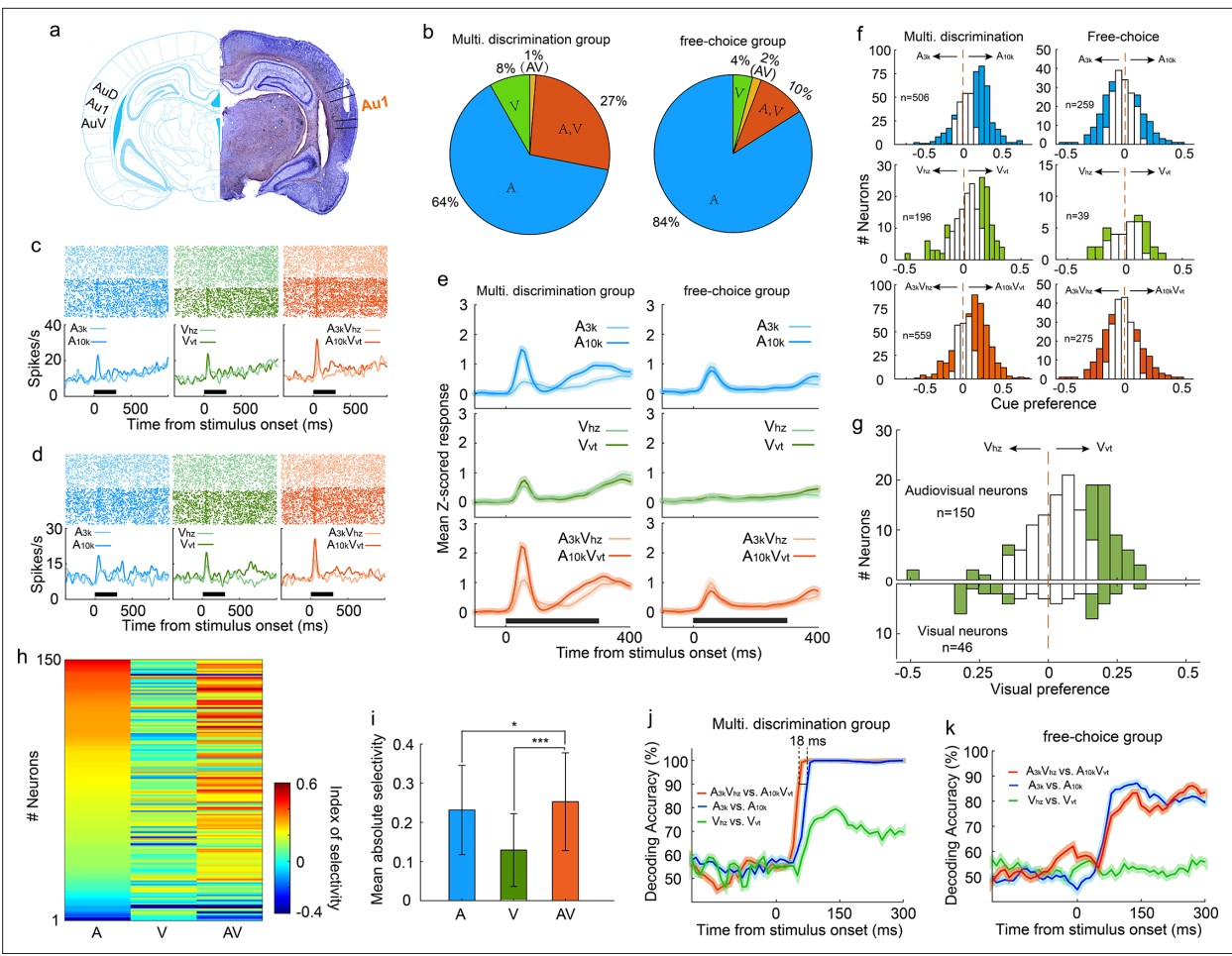

**Figure 2.** Auditory, visual and multisensory selectivity in task engagement. (**a**) Histological verification of recording locations within the primary auditory cortex (Au1). (**b**) Proportion of neurons categorized based on their responsiveness to auditory-only (A), visual-only (V), both auditory and visual (A, V), or audiovisual-only (AV) stimuli during task engagement. (**c–d**) Rasters (top) and peristimulus time histograms (PSTHs, bottom) show responses of two exemplar neurons in $A_{3k}$ (light blue), $A_{10k}$ (dark blue), $V_{hz}$ (light green), $V_{vt}$ (dark green), $A_{3k}V_{hz}$ (light orange), and $A_{10k}V_{vt}$ (dark orange) trials (correct only). Mean spike counts in PSTHs were computed in 10 ms time windows and then smoothed with a Gaussian kernel (σ=50ms). Black bars indicate the stimulus onset and duration. (**e**) Mean normalized response PSTHs across neurons in auditory (top), visual (middle), and multisensory (bottom) trials (correct only) for multisensory discrimination (left) and free-choice (right) groups. Shaded areas represent mean ± SEM. Black bars indicate stimulus onset and duration. (**f**) Histograms of auditory (top), visual (middle) and multisensory (bottom) selectivity for multisensory discrimination (left) and free-choice (right) groups. Filled bars indicate neurons for which the selectivity index was significantly different from 0 (permutation test, p<0.05, bootstrap n=5000). The dash line represents zero. (**g**) Comparison of visual selectivity distribution between audiovisual (top) and visual (bottom) neurons. (**h**) Comparison of auditory (A), visual (V), and multisensory (AV) selectivity of 150 audiovisual neurons, ordered by auditory selectivity. (**i**) Average absolute auditory, visual and multisensory selectivity across 150 audiovisual neurons. Error bars represent SDs. *, p<0.05; ***, p<0.001. (**j**) Decoding accuracy of populations in the multisensory discrimination group. Decoding accuracy (cross-validation accuracy based on SVM) of populations between responses in two auditory (blue), two visual (green), and two multisensory (red) trials. Each decoding value was calculated in a 100ms temporal window moving at the step of 10ms. Shadowing represents the mean ± SD from bootstrapping of 100 repeats. Two dashed lines represent 90% of decoding accuracy for auditory and multisensory conditions. (**k**) Decoding accuracy of populations in the free-choice group.

The online version of this article includes the following figure supplement(s) for figure 2:

**Figure supplement 1.** Characteristic frequency (CF) and response to target sound stimuli of AC neurons recorded in well-trained rats under anesthesia.

**Figure supplement 2.** Stimulus-evoked neural responses and latency comparison for auditory and visual stimuli.

**Figure supplement 3.** Cue preference and multisensory integration of AC neurons in well-trained rats under anesthesia.

**Figure supplement 4.** An auditory cortical neuron exhibiting responsiveness to visual targets in the multisensory discrimination task.

perceptual learning. Notably, such patterns were absent in neurons recorded from a separate group of rats performing free-choice tasks (*Figure 2e*). Further supporting this conclusion is the difference in visual preference between audiovisual and exclusively visual neurons (*Figure 2g*). Among audiovisual neurons with a significant visual selectivity, the majority favored $V_{vt}$ ($V_{vt}$ preferred vs. $V_{hz}$ preferred, 48 vs 7; *Figure 2g*), aligning with their established auditory selectivity. In contrast, visual neurons did not exhibit this bias (12 preferred $V_{vt}$ vs. 13 preferred $V_{hz}$; *Figure 2g*). We propose that the auditory input, which dominates within the auditory cortex, acts as a 'teaching signal' that shapes visual processing through the selective reinforcement of specific visual pathways during associative learning. This aligns with Hebbian plasticity, where stronger auditory responses boost the corresponding visual input, ultimately leading visual selectivity to mirror auditory preference.

Similar to auditory selectivity, the vast majority of neurons (79%, 270/340) showing significant multisensory selectivity exhibited a preference for the multisensory cue ($A_{10k}V_{vt}$) guiding the contralateral choice (*Figure 2f*). To more clearly highlight the influence of visual input on auditory selectivity, we compared auditory, visual, and multisensory selectivity in 150 audiovisual neurons (*Figure 2h*). We found that pairing auditory cues with visual cues significantly improved the neurons' ability to distinguish between auditory stimuli alone (mean absolute auditory selectivity: 0.23±0.11; mean absolute multisensory selectivity: 0.25±0.12; p<0.0001, Wilcoxon Signed Rank Test; *Figure 2i*).

Our multichannel recordings enabled us to decode sensory information from a pseudo-population of AC neurons on a single-trial basis. Using cross-validated support vector machine (SVM) classifiers, we examined how this pseudo-population discriminates stimulus identity within the same modality (e.g., $A_{3k}$ vs. $A_{10k}$ for auditory stimuli, $V_{hz}$ vs. $V_{vt}$ for visual stimuli, $A_{3k}V_{hz}$ vs. $A_{10k}V_{vt}$ for multisensory stimuli). While decoding accuracy was similar for auditory and multisensory conditions, the presence of visual cues accelerated the decoding process, with AC neurons reaching 90% accuracy approximately 18ms earlier in multisensory trials (*Figure 2j*), aligning with behavioral data. Interestingly, AC neurons could discriminate between two visual targets with around 80% accuracy (*Figure 2j*), demonstrating a meaningful incorporation of visual information into auditory cortical processing. However, AC neurons in the free-choice group lacked visual discrimination ability and showed lower accuracy for auditory cues (*Figure 2k*), suggesting that actively engaging multiple senses is crucial for the benefits of multisensory integration.

## Audiovisual integration of AC neurons during the multisensory discrimination task

To understand how AC neurons integrate auditory and visual inputs during task engagement, we compared the multisensory response of each neuron to its strongest corresponding unisensory response using ROC analysis to quantify the difference, termed 'multisensory interactive index' (MSI), where an index value greater than 0 indicates a stronger multisensory response. Over a third (34%, 192 of 559) of AC neurons displayed significant visual modulation of their auditory responses in one or both audiovisual pairings (p<0.05, permutation test), including some neurons with no detectable visual response (104/356). *Figure 3a* exemplifies this, where the multisensory response exceeded the auditory response, a phenomenon termed 'multisensory enhancement' (*Stein and Meredith, 1993*).

Interestingly, AC neurons adopted distinct integration strategies depending on the specific auditory-visual pairing presented. Neurons often displayed multisensory enhancement for one pairing but not another (*Figure 3b*), or even exhibited multisensory inhibition (*Figure 3c*). This enhancement was mainly specific to the $A_{10k}$-$V_{vt}$ pairing, as shown by population averages (*Figure 3d*). Within this pairing, significantly more neurons exhibited enhancement than inhibition (114 vs 36) (*Figure 3e*). In contrast, the $A_{3k}$-$V_{hz}$ pairing showed a balanced distribution of enhancement and inhibition (35 vs 33; *Figure 3e*). This resulted in a significantly higher mean MSI for the $A_{10k}$-$V_{vt}$ pairing compared to the $A_{3k}$-$V_{hz}$ pairing (0.047±0.124 vs 0.003±0.096; paired t-test, p<0.001). Among audiovisual neurons, this biasing is even more pronounced (enhanced vs. inhibited: 62 vs 2 in $A_{10k}$-$V_{vt}$ pairing, 6 vs 13 in $A_{3k}$-$V_{hz}$ pairing; mean MSI: 0.119±0.105 in $A_{10k}$-$V_{vt}$ pairing vs. 0.020±0.083 $A_{3k}$-$V_{hz}$ pairing, paired t-test, p<0.00001; *Figure 3f*). Unlike the early period (0–150ms after cue onset), no significant differences in multisensory integration were observed during the late period (151–300ms after cue onset; *Figure 3—figure supplement 1*).

A similar pattern, albeit less strongly expressed, was observed under anesthesia (*Figure 2—figure supplement 3*), suggesting that multisensory perceptual learning may induce plastic changes in AC.

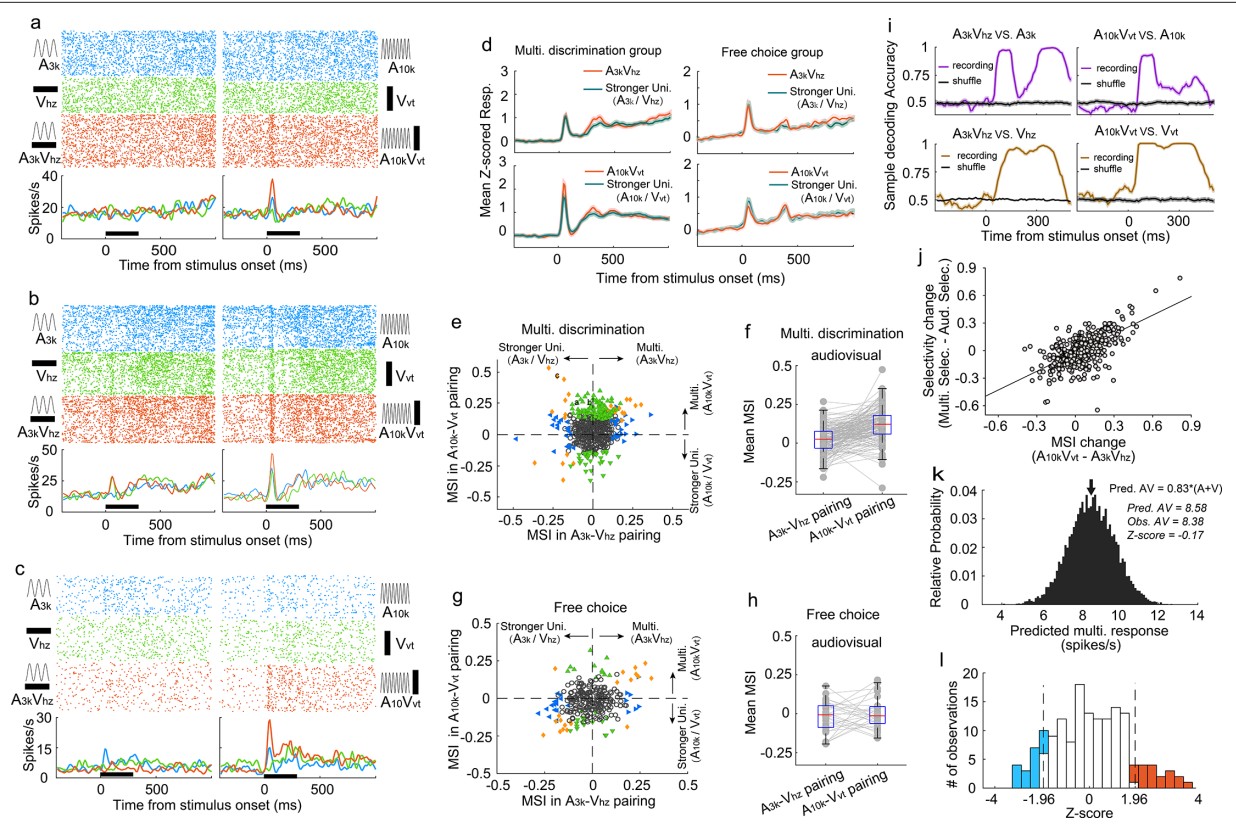

**Figure 3.** Auditory and visual integration in the multisensory discrimination task. (**a–c**) Rasters and PSTHs showing responses of three typical neurons recorded in well-trained rats performing the multisensory discrimination task. (**d**) Population-averaged multisensory responses (red) compared to the corresponding stronger unisensory responses (dark blue) in $A_{3k}V_{hz}$ (top) and $A_{10k}V_{vt}$ (bottom) pairings for multisensory discrimination (left) and free-choice (right) groups. (**e**) Comparison of MSI between $A_{3k}V_{hz}$ (x-axis) and $A_{10k}V_{vt}$ (y-axis) conditions. Each point represents values for a single neuron. Open circles: MSI in either condition was not significant (p>0.05, permutation test); triangles: significant selectivity in either $A_{3k}$-$V_{hz}$ (blue) or $A_{10k}$-$V_{vt}$ (green) condition; Diamonds: significant selectivity in both conditions. Dashed lines show zero MSI. Points labeled a-c correspond to the neurons in panels a–c. (**f**) Mean MSI for $A_{10k}$-$V_{vt}$ and $A_{3k}$-$V_{hz}$ pairings across audiovisual neurons in the multisensory discrimination group. (**g**) Comparison of MSI for the free-choice group. (**h**) Mean MSI across audiovisual neurons for the free-choice group. (**i**) SVM decoding accuracy in AC neurons between responses in multisensory vs. corresponding unisensory trials. Black line indicates shuffled control. (**j**) Positive relationship between the change in MSI ($A_{10k}V_{vt}$ - $A_{3k}V_{hz}$) and the change in selectivity (multisensory selectivity - auditory selectivity). (**k**) Probability density functions of predicted mean multisensory responses (predicted AV) based on 0.83 times the sum of auditory (A) and visual (V) responses (same neuron as in *Figure 2c*). The observed multisensory response matches the predicted mean (Z-score=–0.17). (**l**) Frequency distributions of Z-scores. Open bars indicate no significant difference between actual and predicted multisensory responses. Red bars: Z-scores ≥ 1.96; blue bars: Z-scores ≤ −1.96.

The online version of this article includes the following figure supplement(s) for figure 3:

**Figure supplement 1.** Cue selectivity and multisensory integration in the late period (151–300ms) after cue onset.

**Figure supplement 2.** The observed versus the predicted multisensory response.

In contrast, AC neurons in the free-choice group did not exhibit differential integration patterns (*Figure 3g and h*), indicating that multisensory discrimination and subsequent behavioral reporting are necessary for biased enhancement.

To understand how these distinct integration strategies influence multisensory discrimination, we compared the difference in MSI between $A_{10k}$-$V_{vt}$ and $A_{3k}$-$V_{hz}$ pairings, and the change in neuronal multisensory versus auditory selectivity (*Figure 3j*). We found a significant correlation between these measures ($R$=0.65, p<0.001, Pearson correlation test). The greater the difference in MSI, the more pronounced the increase in cue discrimination during multisensory trials compared to auditory trials. This highlights how distinct integration models enhance cue selectivity in multisensory conditions. Additionally, a subset of neurons exhibited multisensory enhancement for each pairing (*Figure 3e*), potentially aiding in differentiating between multisensory and unisensory responses. SVM decoding

analysis confirmed that population neurons could effectively discriminate between multisensory and unisensory stimuli (*Figure 3i*).

We further explored the integrative mechanisms—whether subadditive, additive, or superadditive—used by AC neurons to combine auditory and visual inputs. Using the bootstrap method, we generated a distribution of predicted multisensory responses by summing mean visual and auditory responses from randomly sampled trials where each stimulus was presented alone. Robust auditory and visual responses were primarily observed in correct contralateral choice trials, so our calculations focused on the $A_{10k}$-$V_{vt}$ pairing. We found that, for most neurons, the observed multisensory response in contralateral choice trials was below the anticipated sum of the corresponding visual and auditory responses (*Figure 3—figure supplement 2*). Specifically, as exemplified in *Figure 3k*, the observed multisensory response approximated 83% of the sum of the auditory and visual responses in most cases, as quantified in *Figure 3l*.

## Impact of incorrect choices on audiovisual integration

To investigate how incorrect choices affected audiovisual integration, we compared multisensory integration for correct and incorrect choices within each auditory-visual pairing, focusing on neurons with a minimum of nine trials per choice per cue. Our findings demonstrated a significant reduction in the magnitude of multisensory enhancement during incorrect choice trials in the $A_{10k}$-$V_{vt}$ pairing. *Figure 4a* illustrates a representative case. The mean MSI for incorrect choices was significantly lower than that for correct choices (correct vs. incorrect: 0.059±0.137 vs 0.006±0.207; p=0.005, paired t-test, *Figure 4b and c*). In contrast, the $A_{3k}$-$V_{hz}$ pairing showed no difference in MSI between correct and incorrect trials (correct vs. incorrect: 0.011±0.081 vs 0.003±0.199; p=0.542, paired t-test, *Figure 4d and e*). Interestingly, correct choices here correspond to ipsilateral behavioral selection, while incorrect choices correspond to contralateral behavioral selection. This indicates that contralateral behavioral choice alone does not guarantee stronger multisensory enhancement. Overall, these findings suggest that the multisensory perception reflected by behavioral choices (correct vs. incorrect) might be shaped by the underlying integration strength. Furthermore, our analysis revealed that incorrect choices were associated with a decline in cue selectivity, as shown in *Figure 4—figure supplement 1*.

## Impact of informational match on audiovisual integration

A pivotal factor influencing multisensory integration is the precise alignment of informational content conveyed by distinct sensory cues. To explore the impact of the association status between auditory and visual cues on multisensory integration, we introduced two new multisensory cues ($A_{10k}V_{hz}$ and $A_{3k}V_{vt}$) into the target cue pool during task engagement. These cues were termed 'unmatched multisensory cues' as their auditory and visual components indicated different behavioral choices. Rats received water rewards with a 50% chance in either port when an unmatched multisensory cue was triggered. Behavioral analysis revealed that Rats displayed notable confusion in response to unmatched multisensory cues, as evidenced by their inconsistent choice patterns (*Figure 5—figure supplement 1*).

We recorded from 280 AC neurons in well-trained rats performing a multisensory discrimination task with matched and unmatched audiovisual cues. We analyzed the integrative and discriminative characteristics of these neurons for matched and unmatched auditory-visual pairings. The results revealed distinct integrative patterns in AC neurons when an auditory target cue was paired with the matched visual cue as opposed to the unmatched visual cue. Neurons typically showed a robust response to $A_{10k}$, which was enhanced by the matched visual cue but was either unaffected (*Figure 5a*) or inhibited (*Figure 5b*) by the unmatched visual cue. In some neurons, both matched and unmatched visual cues enhanced the auditory response but matched cues provided a greater enhancement (*Figure 5c*). We compared the MSI for different auditory-visual pairings (*Figure 5d*). Unlike $V_{vt}$, the unmatched visual cue, $V_{hz}$, generally failed to significantly enhance the response to the preferred sound ($A_{10k}$) in most cases (mean MSI: 0.052±0.097 for $A_{10k}$-$V_{vt}$ pairing vs. 0.016±0.123 for $A_{10k}$-$V_{hz}$ pairing, p<0.0001, paired t-test). This suggests that consistent information across modalities strengthens multisensory integration. In contrast, neither associative nor non-associative visual cue could boost neurons' response to the nonpreferred sound ($A_{3k}$) overall (mean MSI: 0.003±0.008 for $A_{3k}$-$V_{hz}$ pairing vs. 0.003±0.006 for $A_{3k}$-$V_{vt}$ pairing, p=0.19, paired t-test).

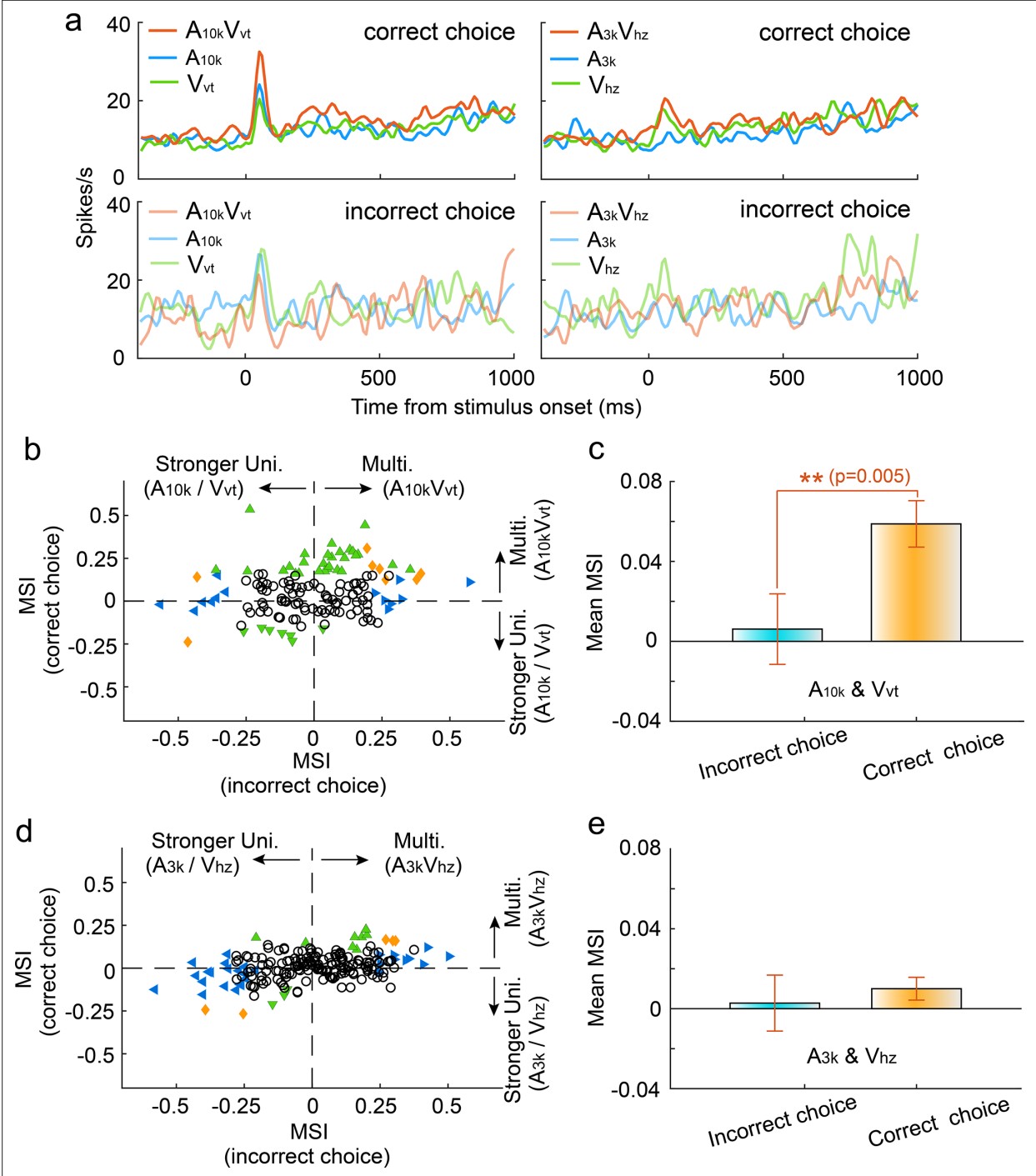

**Figure 4.** Impact of choice selection on audiovisual Integration. (**a**) PSTHs show mean responses of an exemplar neuron for different cue and choice trials. (**b**) Mean MSI across neurons for correct (orange) and incorrect (blue) choices in the $A_{10k}$-$V_{vt}$ pairing. (**c**) Comparison of MSI between correct and incorrect choices for the $A_{10k}$-$V_{vt}$ pairing. Error bars, SEM. (**d, e**) Similar comparisons of MSI for correct and incorrect choices in the $A_{3k}$-$V_{hz}$ pairing.

The online version of this article includes the following figure supplement(s) for figure 4:

**Figure supplement 1.** Comparison of cue selectivity between correct and incorrect choice conditions.

Although distinct AC neurons exhibited varying integration profiles for different auditory-visual pairings, we explored whether, as a population, they could distinguish these pairings. We trained a linear classifier to identify each pair of stimuli and employed the classifier to decode stimulus information from the population activity of grouped neurons. The resulting decoding accuracy matrix for

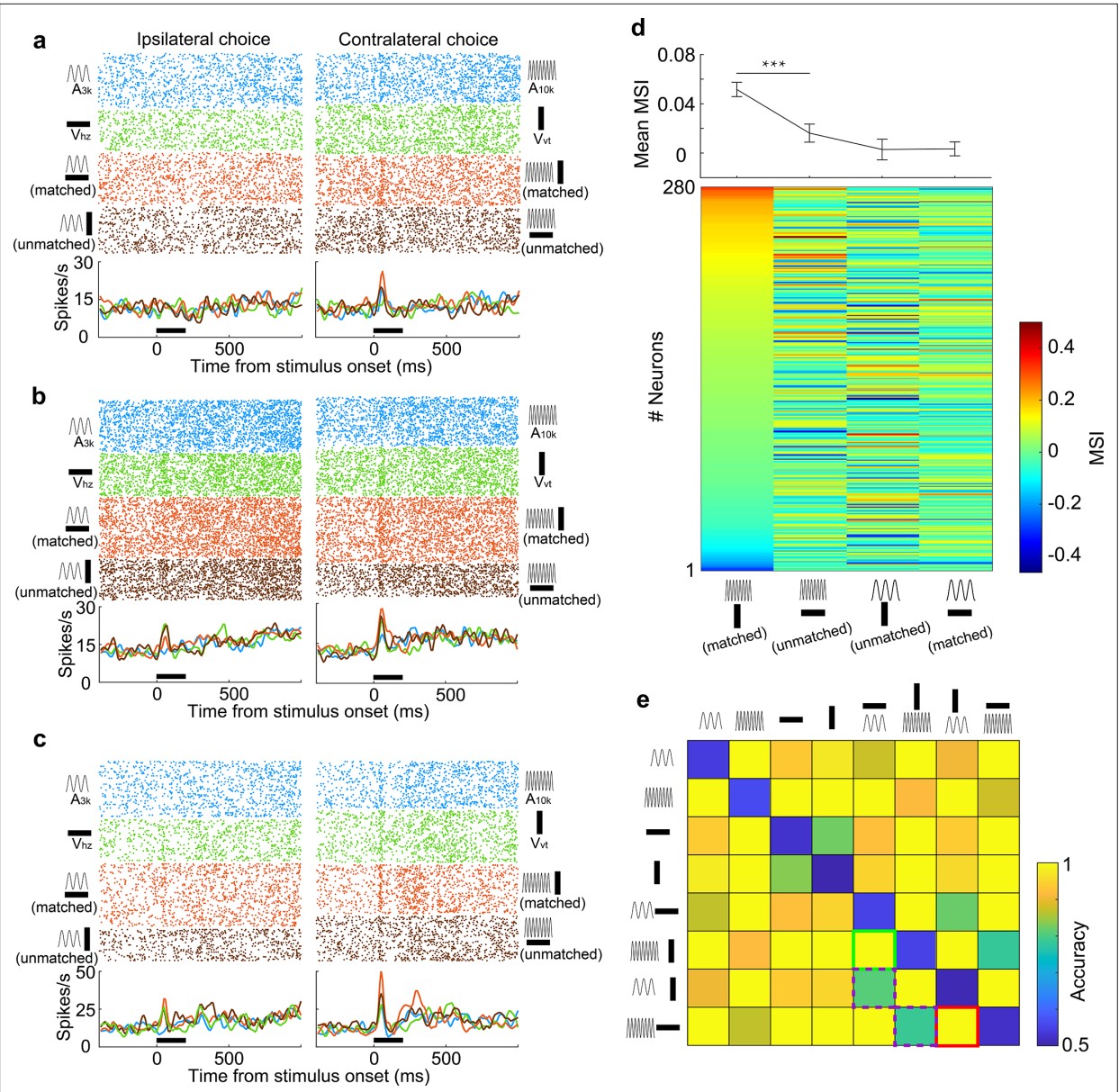

**Figure 5.** Information-dependent integration and discrimination of audiovisual cues. (**a–c**) Responses of three example neurons to six target stimuli and two unmatched multisensory cues ($A_{3k}V_{vt}$ and $A_{10k}V_{hz}$) presented during task engagement. (**d**) Lower panel: MSI for different auditory-visual pairings. Neurons are sorted by their MSI for the $A_{10k}V_{vt}$ condition. Upper panel: Mean MSI across the population for each pairing. Error bar, SEM. ***, p<0.001. (**e**) Population decoding accuracy for all stimulus pairings (8×8) within a 150ms window after stimulus onset. Green, purple, and red squares denote the decoding accuracy for discriminating two target multisensory cues (green), discriminating two unmatched auditory-visual pairings (red) and matched versus unmatched audiovisual pairings (purple), respectively.

The online version of this article includes the following figure supplement(s) for figure 5:

**Figure supplement 1.** Behavioral choice in incongruent and congruent audiovisual trials.

discriminating pairs of stimuli was visualized (***Figure 5e***). We found that the population of neurons could effectively discriminate two target multisensory cues. While matched and unmatched visual cues failed to differentially modulate the response to the nonpreferred sound ($A_{3k}$) at the single-neuron level, grouped neurons still could discriminate them with a decoding accuracy of 0.79 (***Figure 5e***), close to the accuracy of 0.81 for discriminating between $A_{10k}$-$V_{vt}$ and $A_{10k}$-$V_{hz}$ pairings. This suggests that associative learning experiences enabled AC neurons to develop multisensory discrimination

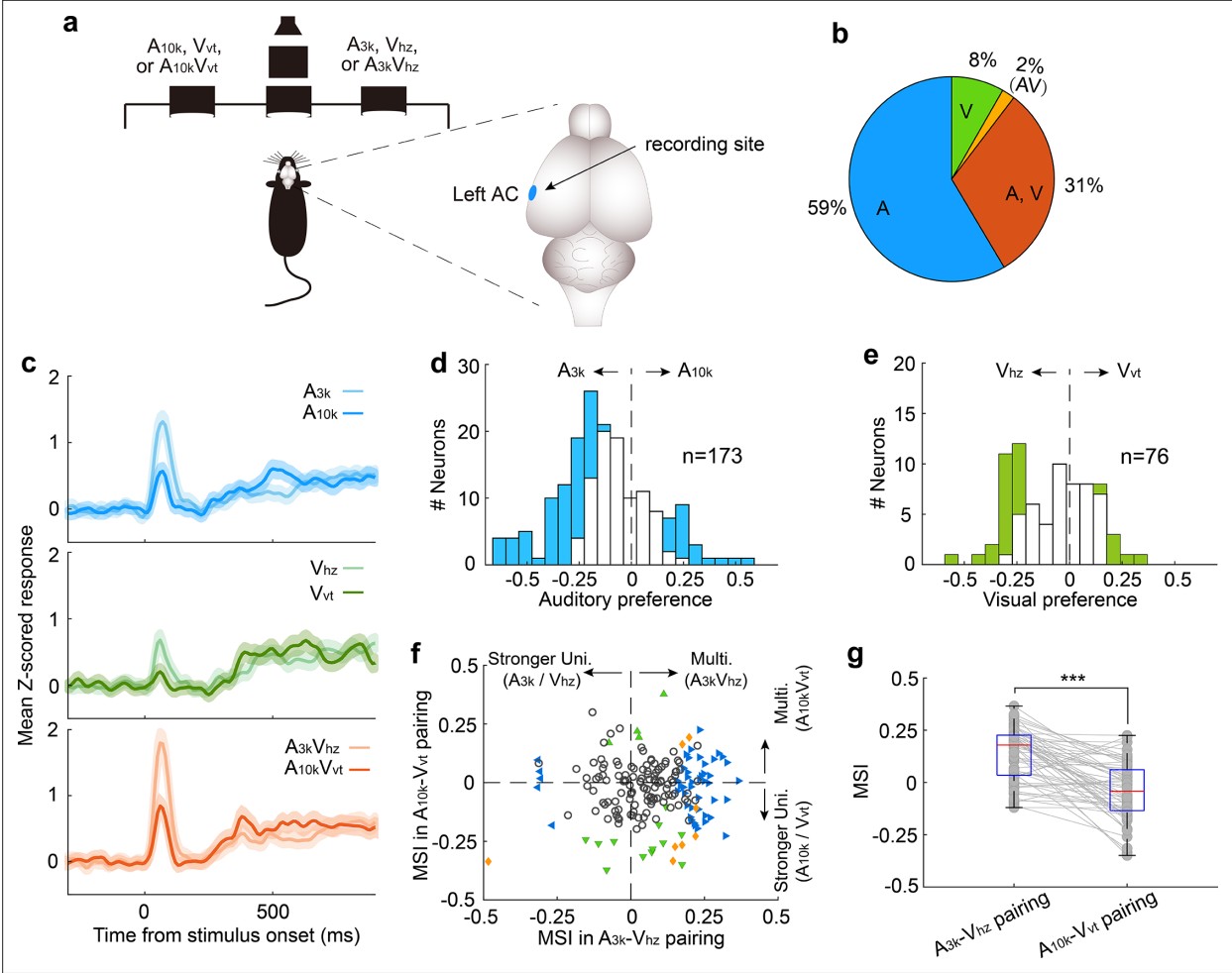

**Figure 6.** Multisensory integration and cue preferences of neurons recorded in left AC. (**a**) The behavioral task and the recording site within the brain (the left AC). (**b**) The proportion of neurons responsive to auditory-only (A), visual-only (V), both auditory and visual (A, V), and audiovisual-only (VA) stimuli based on their spiking activity. (**c**) The mean PSTHs of normalized responses across neurons for different cue conditions. (**d**) A histogram of auditory selectivity index. Filled bars represent neurons with a significant selectivity index (permutation test, p<0.05, bootstrap n=5000). (**e**) Similar to panel (**d**), this panel shows a histogram of the visual selectivity index. (**f**) Comparison of MSI between A₃ₖ-Vₕz and A₁₀ₖ-Vᵥt pairings. (**g**) A boxplot shows the comparison of MSI for audiovisual neurons. ***, p<0.001. The conventions used are consistent with those in *Figures 2 and 3*.

abilities. However, the accuracy for discriminating matched vs. unmatched cues was lower compared to other pairings (*Figure 5e*).

## Cue preference and multisensory integration in left AC

Our data showed that most neurons in the right AC preferred the cue directing the contralateral choice, regardless of whether it was auditory or visual. However, this could simply be because these neurons were naturally more responsive to those specific cues, not necessarily because they learned an association between the cues and the choice. To address this, we trained another four rats to perform the same discrimination task but recorded neuronal activity in left AC (*Figure 6a*). We analyzed 193 neurons, of which about a third (31%, 60/193) responded to both auditory and visual cues (*Figure 6b*). Similar to the right AC, the average response across neurons in the left AC preferred cues guiding the contralateral choice (*Figure 6c*). However, in this case, the preferred cues are A₃ₖ, Vₕz and the audiovisual pairing A₃ₖVₕz. As shown in *Figure 6d*, more auditory-responsive neurons favored the sound denoting the contralateral choice. The same was true for visual selectivity in visually responsive neurons (*Figure 6e*). This strongly suggests that the preference was not simply due to a general bias toward a specific cue, but rather reflected the specific associations the animals learned during the training.

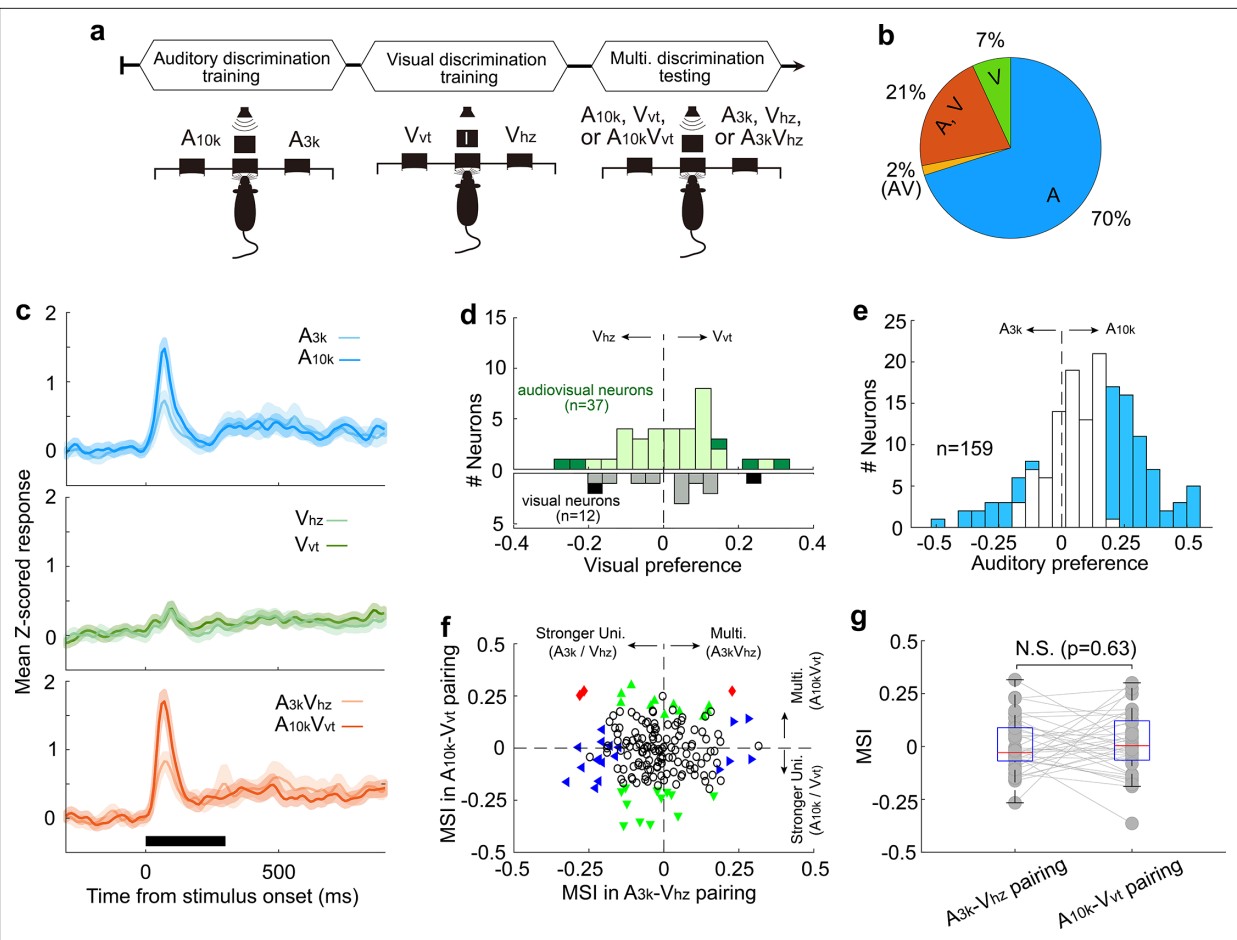

**Figure 7.** Multisensory integration and cue preferences in right AC neurons after unisensory training. (**a**) This panel depicts the training stages for the rats: auditory discrimination training followed by visual discrimination training. (**b**) The proportion of neurons of different types. (**c**) Mean normalized responses across neurons for different cue conditions. (**d, e**) Histograms of visual (**d**) and auditory (**e**) selectivity. (**f**) Comparison of neuronal MSI between $A_{3k}$-$V_{hz}$ and $A_{10k}$-$V_{vt}$ pairings. (**g**) MSI comparison in audiovisual neurons. N.S., no significant difference. The conventions used are consistent with those in **Figures 2 and 3** and **Figure 6**.

Consistent with the cue biasing, differential multisensory integration was observed, with multisensory enhancement biased toward the $A_{3k}$-$V_{hz}$ pairing (**Figure 6f**). This mirrors the finding in the right AC, where multisensory enhancement was biased toward the auditory-visual pairing guiding the contralateral choice. In audiovisual neurons, mean MSI in the $A_{3k}$-$V_{hz}$ pairing is substantially higher than in the $A_{10k}$-$V_{vt}$ pairing (0.135±0.126 to -0.039±0.138; p<0.0001, paired t-test; **Figure 6g**). These findings suggest that AC neurons exhibit cue preference and biased multisensory enhancement based on the learned association between cues and behavioral choice. This mechanism could enable the brain to integrate cues of different modalities into a common behavioral response.

## Unisensory training does not replicate multisensory training effects

Our data suggest that most AC audiovisual neurons exhibited a visual preference consistent with their auditory preference following consistent multisensory discrimination training. Additionally, these neurons developed selective multisensory enhancement for a specific audiovisual stimulus pairing. To investigate whether these properties stemmed solely from long-term multisensory discrimination training, we trained a new group of animals (n=3) first on auditory and then on visual discriminations (**Figure 7a**). These animals did not receive task-related multisensory associations during the training period. We then examined the response properties of neurons recorded in the right AC when well-trained animals performed auditory, visual and audiovisual discrimination tasks.

Among recorded AC neurons, 28% (49/174) responded to visual target cues (*Figure 7b*). Unlike the multisensory training group, most visually responsive neurons in the unisensory training group lacked a visual preference, regardless of whether they were visually-only responsive or audiovisual (*Figure 7c*). Interestingly, similar to the multisensory training group, nearly half of the recorded neurons (47%, 75/159) demonstrated clear auditory discrimination, with most (80%, 60 out of 75) favoring the sound guiding the contralateral choice (*Figure 7d*). Furthermore, the unisensory training group did not exhibit population-level multisensory enhancement (n=174, *Figure 7f*). The mean MSI for the $A_{3k}$-$V_{Hz}$ and $A_{10k}$-$V_{Hz}$ pairs showed no significant difference (p=0.327, paired t-test), with values of –0.02±0.12 and –0.01±0.13, respectively (*Figure 7g*). Even among audiovisual neurons (n=37), multisensory integration did not differ significantly between the two pairings ($A_{3k}$-$V_{Hz}$ vs $A_{10k}$-$V_{Hz}$: 0.002±0.126 vs 0.017±0.147; p=0.63; *Figure 7g*). These findings suggest that the development of multisensory enhancement for specific audiovisual cues and the alignment of auditory and visual preferences likely depend on the association of auditory and visual stimuli with the corresponding behavioral choice during multisensory training. Unisensory training alone cannot replicate these effects.

## Discussion

In this study, we investigated how AC neurons integrate auditory and visual inputs to discriminate multisensory objects and whether this integration reflects the congruence between auditory and visual cues. Our findings reveal that most AC neurons exhibited distinct integrative patterns for different auditory-visual pairings, with multisensory enhancement observed primarily for favored pairings. This indicates that AC neurons show significant selectivity in their integrative processing. Furthermore, AC neurons effectively discriminated between matched and mismatched auditory-visual pairings, highlighting the crucial role of the AC in multisensory object recognition and discrimination. Interestingly, a subset of auditory neurons not only developed visual responses but also exhibited congruence between auditory and visual selectivity. These findings suggest that multisensory perceptual training establishes a memory trace of the trained audiovisual experiences within the AC and enhances the preferential linking of auditory and visual inputs. Sensory cortices, like AC, may act as a vital bridge for communicating sensory information across different modalities.

Numerous studies have explored the cross-modal interaction of sensory cortices (*Perrodin et al., 2015*; *Kayser et al., 2008*; *Bizley et al., 2007*; *Meijer et al., 2017*; *Deneux et al., 2019*; *Ghazanfar and Schroeder, 2006*). Recent research has highlighted the critical role of cross-modal modulation in shaping the stimulus characteristics encoded by sensory cortical neurons. For instance, sound has been shown to refine orientation tuning in layer 2/3 neurons of the primary visual cortex (*Ibrahim et al., 2016*), and a congruent visual stimulus can enhance sound representation in the AC (*Atilgan et al., 2018*). Meijer et al. reported that congruent audiovisual stimuli evoke balanced enhancement and suppression in V1, while incongruent stimuli predominantly lead to suppression (*Meijer et al., 2017*), mirroring our findings in AC, where multisensory integration was dependent on stimulus feature. Despite these findings, the functional role and patterns of cross-modal interaction during perceptual discrimination remain unclear. In this study, we made a noteworthy discovery that AC neurons employed a nonuniform mechanism to integrate visual and auditory stimuli while well-trained rats performed a multisensory discrimination task. This differentially integrative pattern improved multisensory discrimination. Notably, this integrative pattern did not manifest during a free-choice task. These findings indicate that depending on the task, different sensory elements need to be combined to guide adaptive behavior. We propose that task-related differentially integrative patterns may not be exclusive to sensory cortices but could represent a general model in the brain.

Consistent with prior research (*Chang et al., 2022*; *Zheng et al., 2021*), most AC neurons exhibited a selective preference for cues associated with contralateral choices, regardless of the sensory modality. This suggests that AC neurons may contribute to linking sensory inputs with decision-making, although their causal role remains to be examined. Associative learning may drive the formation of new connections between sensory and motor areas of the brain, such as cortico-cortical pathways (*Makino et al., 2016*). Notably, this cue-preference biasing was absent in the free-choice group. A similar bias was also reported in a previous study, where auditory discrimination learning selectively potentiated corticostriatal synapses from neurons representing either high or low frequencies associated with contralateral choices (*Xiong et al., 2015*). These findings highlight the significant role

of associative learning in cue encoding, emphasizing the integration of stimulus discrimination and behavioral choice.

Prior work has investigated how neurons in sensory cortices discriminate unisensory cues in perceptual tasks (*Romo and Rossi-Pool, 2020*; *Juavinett et al., 2018*; *Schoups et al., 2001*). It is well-established that when animals learn the association of sensory features with corresponding behavioral choices, cue representations in early sensory cortices undergo significant changes (*Otazu et al., 2009*; *Yan et al., 2014*; *Xin et al., 2019*; *Makino and Komiyama, 2015*). For instance, visual learning shapes cue-evoked neural responses and increases visual selectivity through changes in the interactions and correlations between visual cortical neurons (*Chadwick et al., 2023*; *Khan et al., 2018*). Consistent with these previous studies, we found that auditory discrimination in the AC of well-trained rats substantially improved during the task.

In this study, we extended our investigation to include multisensory discrimination. We discovered that when rats performed the multisensory discrimination task, AC neurons exhibited robust multisensory selectivity, responding strongly to one auditory-visual pairing and showing weak or negligible responses to the other, similar to their behavior in auditory discrimination. Audiovisual neurons demonstrated higher selectivity in multisensory trials compared to auditory trials, driven by the differential integration pattern. Additionally, more AC neurons were involved in auditory-visual discrimination compared to auditory discrimination alone, suggesting that the recruitment of additional neurons may partially explain the higher behavioral performance observed in multisensory trials. A previous study indicates that cross-modal recruitment of more cortical neurons also enhances perceptual discrimination (*Lomber et al., 2010*).

Our study explored how multisensory discrimination training influences visual processing in the AC. We observed well-trained rats exhibited a higher number of AC neurons responding to visual cues compared to untrained rats. This finding aligns with growing evidence that multisensory perceptual learning can effectively drive plastic change in both sensory and association cortices (*Shams and Seitz, 2008*; *Proulx et al., 2014*). Our results extend prior findings (*Bizley et al., 2007*; *Morrill and Hasenstaub, 2018*), showing that visual input not only reaches the AC but can also drive discriminative responses, particularly during task engagement. This task-specific plasticity enhances cross-modal integration, as demonstrated in other sensory systems. For example, calcium imaging studies in mice showed that a subset of multimodal neurons in visual cortex develops enhanced auditory responses to the paired auditory stimulus following coincident auditory–visual experience (*Knöpfel et al., 2019*). A study conducted on the gustatory cortex of alert rats has shown that cross-modal associative learning was linked to a dramatic increase in the prevalence of neurons responding to nongustatory stimuli (*Vincis and Fontanini, 2016*). Moreover, in the primary visual cortex, experience-dependent interactions can arise from learned sequential associations between auditory and visual stimuli, mediated by corticocortical connections rather than simultaneous audiovisual presentations (*Garner and Keller, 2022*).

Among neurons responding to both auditory and visual stimuli, a congruent visual and auditory preference emerged during multisensory discrimination training, as opposed to unisensory discrimination training. These neurons primarily favored visual cues that matched their preferred sound. Interestingly, this preference was not observed in neurons solely responsive to visual targets. The strength of a visual response seems to be contingent upon the paired auditory input received by the same neuron. This aligns with known mechanisms in other brain regions, where learning strengthens or weakens connections based on experience (*Knöpfel et al., 2019*; *Komiyama et al., 2010*). This synchronized auditory-visual selectivity may be a way for AC to bind corresponding auditory and visual features, potentially forming memory traces for learned multisensory objects. These results indicate that multisensory training could drive the formation of specialized neural circuits within the auditory cortex, facilitating integrated processing of related auditory and visual information. However, further causal studies are required to confirm this hypothesis and to determine whether the auditory cortex is the primary site of these circuit modifications.

There is ongoing debate about whether cross-sensory responses in sensory cortices predominantly reflect sensory inputs or are influenced by behavioral factors, such as cue-induced body movements. A recent study shows that sound-clip evoked activity in visual cortex have a behavioral rather than sensory origin and is related to stereotyped movements (*Bimbard et al., 2023*). Several studies have demonstrated sensory neurons can encode signals associated with whisking (*Stringer et al.,*

2019), running (**Niell and Stryker, 2010**), pupil dilation (**Vinck et al., 2015**) and other movements (**Musall et al., 2019**). In our study, the responses to visual stimuli in the auditory cortex occurred primarily within a 100ms window following cue onset, suggesting that visual information reaches the AC through rapid pathways. Potential candidates include direct or fast cross-modal inputs, such as pulvinar-mediated pathways (**Chou et al., 2020**) or corticocortical connections (**Atilgan et al., 2018**; **Schmehl and Groh, 2021**), rather than slower associative mechanisms. This early timing indicates that the observed responses were less likely modulated by visually evoked body or orofacial movements, which typically occur with a delay relative to sensory cue onset (**Oude Lohuis et al., 2024**).

A recent study by **Clayton et al., 2024** demonstrated that sensory stimuli can evoke rapid motor responses, such as facial twitches, within 50ms, mediated by subcortical pathways and modulated by descending corticofugal input (**Clayton et al., 2024**). These motor responses provide a sensitive behavioral index of auditory processing. Although Clayton et al. did not observe visually evoked facial movements, it is plausible that visually driven motor activity occurs more frequently in freely moving animals compared to head-fixed conditions. In goal-directed tasks, such rapid motor responses might contribute to the contralaterally tuned responses observed in our study, potentially reflecting preparatory motor behaviors associated with learned responses. Consequently, some of the audiovisual integration observed in the auditory cortex may represent a combination of multisensory processing and preparatory motor activity. Comprehensive investigation of these motor influences would require high-speed tracking of orofacial and body movements. Therefore, our findings should be interpreted with this consideration in mind. Future studies should aim to systematically monitor and control eye, orofacial, and body movements to disentangle sensory-driven responses from motor-related contributions, enhancing our understanding of motor planning's role in multisensory integration.

Additionally, our study sheds light on the role of semantic-like information in multisensory integration. During training, we created an association between specific auditory and visual stimuli, as both signaled the same behavioral choice. This setup mimics real-world scenarios where visual and auditory cues possess semantic coherence, such as an image of a cat paired with the sound of a 'meow'. Previous research has shown that semantically congruent multisensory stimuli enhance behavioral performance, while semantically incongruent stimuli either show no enhancement or result in decreased performance (**Laurienti et al., 2004**). Intriguingly, our findings revealed a more nuanced role for semantic information. While AC neurons displayed multisensory enhancement for the preferred congruent audiovisual pairing, this wasn't for enhanced multisensory integration. However, the strength of multisensory enhancement itself served as a key indicator in differentiating between matched and mismatched cues. These findings provide compelling evidence that the nature of the semantic-like information plays a vital role in modifying multisensory integration at the neuronal level.

## Materials and methods

### Animals

The animal procedures conducted in this study (Protocol number: m+R20211004) were ethically approved by the Local Ethical Review Committee of East China Normal University and followed the guidelines outlined in the Guide for the Care and Use of Laboratory Animals of East China Normal University. Twenty-five adult male Long-Evans rats (age: 10–12 weeks), weighing approximately 250 g, were obtained from the Shanghai Laboratory Animal Center (Shanghai, China) and used as subjects for the experiments. Of these rats, 14 were assigned to the multisensory discrimination task experiments, 3 to the unisensory discrimination task, and 8 to the control free-choice task experiments. The rats were group-housed with no more than four rats per cage and maintained on a regular light-dark cycle. They underwent water deprivation for 2 days before the start of behavioral training, with water provided exclusively inside the training box on training days. Training sessions were conducted 6 days per week, each lasting between 50 and 80 min, and were held at approximately the same time each day to minimize potential circadian variations in performance. The body weight of the rats was carefully monitored throughout the study, and supplementary water was provided to those unable to maintain a stable body weight from task-related water rewards.

## Behavioral apparatus

All experiments were conducted in a custom-designed operant chamber, measuring 50×30 × 40 cm (length ×width × height), with an open-top design. The chamber was placed in a sound-insulated double-walled room, and the inside walls and ceiling were covered with 3 inches of sound-absorbing foam to minimize external noise interference. One sidewall of the operant chamber was equipped with three snout ports, each monitored by a photoelectric switch (see *Figure 1a*).

Automated training procedures were controlled using a real-time control program developed in MATLAB (Mathworks, Natick, MA, USA). The auditory signals generated by the program were sent to an analog-digital multifunction card (NI USB 6363, National Instruments, Austin, TX, USA), amplified by a power amplifier (ST-601, SAST, Zhejiang, China), and delivered through a speaker (FS Audio, Zhejiang, China). The auditory stimuli consisted of 300ms-long (15ms ramp-decay) pure tones with a frequency of 3 kHz or 10 kHz. The sound intensity was set at 60 dB sound pressure level (SPL) against an ambient background of 35–40 dB SPL. SPL measurements were taken at the position of the central port, which served as the starting position for the rats.

The visual cue was generated using two custom-made devices located on each side in front of the central port. Each device consisted of two arrays of closely aligned light-emitting diodes arranged in a cross pattern. The light emitted by the vertical or horizontal LED array passed through a ground glass, resulting in the formation of the corresponding vertical or horizontal light bar, each measuring 6×0.8 cm (*Figure 1*). As stimuli, the light bars were illuminated for 300ms at an intensity of 2~3 cd/m². The audiovisual cue (multisensory cue) was the simultaneous presentation of both auditory and visual stimuli.

## Multisensory discrimination task

The rats were trained to perform a cue-guided two-alternative forced-choice task, modified from previously published protocols (*Chang et al., 2022*; *Zheng et al., 2021*). Each trial began with the rat placing its nose into the center port. Following a short variable delay period (500–700ms) after nose entry, a randomly selected stimulus signal was presented. Upon cue presentation, the rats were allowed to initiate their behavioral choice by moving to either the left or right port (*Figure 1a*). The training consisted of two stages. In the first stage, which typically lasted 3–5 weeks, the rats were trained to discriminate between two audiovisual cues. In the second stage, an additional four unisensory cues were introduced, training the rats to discriminate a total of six cues (two auditory, two visual, and two audiovisual). This stage also lasted approximately 3–5 weeks.

During the task, the rats were rewarded with a drop (15–20 μl) of water when they moved to the left reward port following the presentation of a 10 kHz pure tone sound, a vertical light bar, or their combination. For trials involving a 3 kHz tone sound, a horizontal light bar, or their combination, the rats were rewarded when they moved to the right reward port. Incorrect choices or failures to choose within 3 s after cue onset resulted in a timeout punishment of 5–6 s. Typically, the rats completed between 300 and 500 trials per day. They were trained to achieve a competency level of more than 80% correct overall and >70% correct in each cue condition in three consecutive sessions before the surgical implantation of recording electrodes.

The correct rate was calculated as follows:

Correct rate (%)=100*(the number of correct trials) / (total number of trials).

The reaction time was defined as the temporal gap between the cue onset and the time when the rat withdrew its nose from the infrared beam in the cue port.

## Unisensory discrimination task

The rats first learned to discriminate between two auditory cues. Once their performance in the auditory discrimination task exceeded 75% correct, the rats then learned to discriminate between two visual cues. The auditory and visual cues used were the same as those in the multisensory discrimination task.

## No cue discrimination free-choice task

In this task, the rats were not required to discriminate the cues presented. They received a water reward at either port following the onset of the cue, and their port choice was spontaneous. The six

cues used were the same as those in the multisensory discrimination task. One week of training was sufficient for the rats to learn this no cue discrimination free-choice task.

## Assembly of tetrodes

The tetrodes were constructed using Formvar-Insulated Nichrome Wire (bare diameter: 17.78 μm, A-M systems, WA, USA) twisted in groups of four. Two 20-cm-long wires were folded in half over a horizontal bar to facilitate twisting. The ends of the wires were clamped together and manually twisted in a clockwise direction. The insulation coating of the twisted wires was then fused using a heat gun at the desired twist level. Subsequently, the twisted wire was cut in the middle to produce two tetrodes. To enhance the longitudinal stability of each tetrode, it was inserted into polymide tubing (inner diameter: 0.045 inches; wall: 0.005 inches; A-M systems, WA, USA) and secured in place using cyanoacrylate glue. An array of 2×4 tetrodes was assembled, with an inter-tetrode gap of 0.4–0.5 mm. After assembly, the insulation coating at the tip of each wire was gently removed, and the exposed wire was soldered to a connector pin. For the reference electrode, a Ni-Chrome wire with a diameter of 50.8 μm (A-M systems, WA, USA) was used, with its tip exposed. A piece of copper wire with a diameter of 0.1 mm served as the ground electrode. Each of these electrodes was also soldered to a corresponding pin on the connector. The tetrodes, reference electrode, and ground electrode, along with their respective pins, were carefully arranged and secured using silicone gel. Immediately before implantation, the tetrodes were trimmed to an appropriate length.

## Electrode implantation

Prior to surgery, the animal received subcutaneous injections of atropine sulfate (0.01 mg/kg) to reduce bronchial secretions. The animal was then anesthetized with an intraperitoneal (i.p.) injection of sodium pentobarbital (40–50 mg/kg) and securely positioned on a stereotaxic apparatus (RWD, Shenzhen, China). An incision was made in the scalp, and the temporal muscle was carefully recessed. Subsequently, a craniotomy and durotomy were performed to expose the target brain region. Using a micromanipulator (RWD, Shenzhen, China), the tetrode array was precisely positioned at stereotaxic coordinates 3.5–5.5 mm posterior to bregma and 6.4 mm lateral to the midline, and advanced to a depth of approximately 2–2.8 mm from the brain surface, corresponding to the primary auditory cortex. The craniotomy was then sealed with tissue gel (3 M, Maplewood, MN, USA). The tetrode array was secured to the skull using stainless steel screws and dental acrylic. Following the surgery, animals received a 4-day prophylactic course of antibiotics (Baytril, 5 mg/kg, body weight, Bayer, Whippany, NJ, USA). They were allowed a recovery period of at least 7 days (typically 9–12 days) with free access to food and water.

## Neural recordings and analysis

After rats had sufficiently recovered from the surgery, they resumed performing the same behavioral task they were trained to accomplish before surgery. Recording sessions were initiated once the animals' behavioral performance had returned to the level achieved before the surgery (typically within 2–3 days). Wideband neural signals in the range of 300–6000 Hz were recorded using the AlphaOmega system (AlphaOmega Instruments, Nazareth Illit, Israel). The amplified signals (×20) were digitized at a sampling rate of 25 kHz. These neural signals, along with trace signals representing the stimuli and session performance information, were transmitted to a PC for online observation and data storage. Neural responses were analyzed within a 0–150ms temporal window after cue onset, as this period was identified as containing the main cue-evoked responses for most neurons. This time window was selected based on the consistent and robust neural activity observed during this period.

Additionally, we recorded neural responses from well-trained rats under anesthesia. Anesthesia was induced with an intraperitoneal injection of sodium pentobarbital (40 mg/kg body weight) and maintained throughout the experiment by continuous intraperitoneal infusion of sodium pentobarbital (0.004 ~ 0.008 g/kg/hr) using an automatic microinfusion pump (WZ-50C6, Smiths Medical, Norwell, MA, USA). The anesthetized rats were placed in the behavioral training chamber, and their heads were positioned in the cue port to mimic the cue-triggering as observed during task engagement. To maintain body temperature, a heating blanket was used to maintain a temperature of 37.5 °C. The same auditory, visual, and audiovisual stimuli used during task engagement were randomly presented to the anesthetized rats. For each cue condition, we recorded 40–60 trials of neural responses.

## Analysis of electrophysiological data

The raw neural signals were recorded and saved for subsequent offline analysis. Spike sorting was performed using Spike 2 software (CED version 8, Cambridge, UK). Initially, the recorded raw neural signals were band-pass filtered in the range of 300–6000 Hz to eliminate field potentials. A threshold criterion, set at no less than three times the standard deviation (SD) above the background noise, was applied to automatically identify spike peaks. The detected spike waveforms were then subjected to clustering using template-matching and built-in principal component analysis tool in a three-dimensional feature space. Manual curation was conducted to refine the sorting process. Each putative single unit was evaluated based on its waveform and firing patterns over time. Waveforms with inter-spike intervals of less than 2.0ms were excluded from further analysis. Spike trains corresponding to an individual unit were aligned to the onset of the stimulus and grouped based on different cue and choice conditions. Units were included in further analysis only if their presence was stable throughout the session, and their mean firing rate exceeded 2 Hz. The reliability of auditory and visual responses for each unit was assessed, with well-isolated units typically showing the highest response reliability. To generate peristimulus time histograms (PSTHs), the spike trains were binned at a resolution of 10ms, and the average firing rate in each bin was calculated. The resulting firing rate profile was then smoothed using a Gaussian kernel with a standard deviation ($\sigma$) of 50ms. In order to normalize the firing rate, the mean firing rate and SD during a baseline period (a 400 ms window preceding cue onset) were used to convert the averaged firing rate of each time bin into a Z-score.

## Population decoding

To evaluate population discrimination for paired stimuli (cue_A vs. cue_B), we trained an SVM classifier with a linear kernel to predict cue selectivity. The SVM classifier was implemented using the 'fitcsvm' function in Matlab. In this analysis, spike counts for each neuron in correct trials were grouped based on the triggered cues and binned into a 100ms window with a 10ms resolution. To minimize overfitting, only neurons with more than 30 trials for each cue were included. All these neurons were combined to form a pseudo population. The responses of the population neurons were organized into an M × N × T matrix, where M is the number of trials, N is the number of neurons, and T is the number of bins. For each iteration, 30 trials were randomly selected for each cue from each neuron. During cross-validation, 90% of the trials were randomly sampled as the training set, while the remaining 10% were used as the test set. The training set was used to compute the linear hyperplane that optimally separated the population response vectors corresponding to cue_A vs cue_B trials. The performance of the classifier was calculated as the fraction of correctly classified test trials, using 10-fold cross-validation procedures. To ensure robustness, we repeated the resampling process 100 times and computed the mean and standard deviation of the decoding accuracy across the 100 resampling iterations. Decoders were trained and tested independently for each bin. To assess the significance of decoding accuracy exceeding the chance level, shuffled decoding procedures were conducted by randomly shuffling the trial labels for 1000 iterations.

## Cue selectivity

To quantify cue (auditory, visual and multisensory) selectivity between two different cue conditions (e.g. low tone trials vs. high tone trials), we employed a ROC-based analysis, following the method described in a previous study (*Britten et al., 1992*). This approach allowed us to assess the difference between responses in cue_A and cue_B trials. Firstly, we established 12 threshold levels of neural activity that covered the range of firing rates obtained in both cue_A and cue_B trials. For each threshold criterion, we plotted the proportion of cue_A trials where the neural response exceeded the criterion against the proportion of cue_B trials exceeding the same criterion. This process generated an ROC curve, from which we calculated the area under the ROC curve (auROC). The cue selectivity value was then defined as 2 * (auROC - 0.5). A cue selectivity value of 0 indicated no difference in the distribution of neural responses between cue_A and cue_B, signifying similar responsiveness to both cues. Conversely, a value of 1 or –1 represented the highest selectivity, indicating that responses triggered by cue_A were consistently higher or lower than those evoked by cue_B, respectively. To determine the statistical significance of the observed cue selectivity, we conducted a two-tailed permutation test with 2000 permutations. We randomly reassigned the neural responses to cue_A and cue_B trials and recalculated a cue selectivity value for each permutation. This generated a distribution of values from

which we calculated the probability of our observed result. If the observed ROC value exceeds the top 2.5% of the distribution or falls below the bottom 2.5%, it was deemed significant (i.e. p<0.05).

## Comparison of actual and predicted multisensory responses

We conducted a comprehensive analysis to compare the observed multisensory responses with predicted values. The predicted multisensory response is calculated as either the sum of visual and auditory responses or as a coefficient multiplied by this sum. To achieve this, we first computed the mean observed multisensory response by averaging across audiovisual trials. We then created a benchmark distribution of predicted multisensory responses by iteratively calculating all possible predictions. In each iteration, we randomly selected (without replacement) the same number of trials as used in the actual experiment for both auditory and visual conditions. The responses from these selected auditory and visual trials were then averaged to obtain mean predicted auditory and visual responses, which were used to create a predicted multisensory response. This process was repeated 5000 times to generate a comprehensive reference distribution of predicted multisensory responses. By comparing the actual mean multisensory response to this distribution, we expressed their relationship as a Z-score. This method allowed us to quantitatively assess multisensory interactions, providing valuable insights into neural processing and enhancing our understanding of the mechanisms underlying multisensory integration.

## Histology

Following the final data recording session, the precise tip position of the recording electrode was marked by creating a small DC lesion (–30 µA for 15 s). Subsequently, the rats were deeply anesthetized with sodium pentobarbital (100 mg/kg) and underwent transcardial perfusion with saline for several minutes, followed immediately by PBS containing 4% paraformaldehyde (PFA). The brains were carefully extracted and immersed in the 4% PFA solution overnight. To ensure optimal tissue preservation, the fixed brain tissue underwent cryoprotection in PBS with a 20% sucrose solution for at least 3 days. Afterward, the brain tissue was coronally sectioned using a freezing microtome (Leica, Wetzlar, Germany) with a slice thickness of 50 µm. The resulting sections, which contained the auditory cortex, were stained with methyl violet to verify the lesion sites and/or the trace of electrode insertion within the primary auditory cortex.

## Statistical analysis

We also use ROC analysis to calculate the MSI to denote the difference between multisensory and the corresponding stronger unisensory responses. All statistical analyses were conducted in MATLAB, and statistical significance was defined as a p value of<0.05. To determine the responsiveness of AC neurons to sensory stimuli, neurons exhibiting evoked responses greater than 2 spikes/s within a 0.3 s window after stimulus onset, and significantly higher than the baseline response (p < 0.05, Wilcoxon signed-rank test), were included in the subsequent analysis. For behavioral data, such as mean reaction time differences between unisensory and multisensory trials, cue selectivity and mean MSI across different auditory-visual conditions, comparisons were performed using either the paired t-test or the Wilcoxon signed-rank test. The Shapiro-Wilk test was conducted to assess normality, with the paired t-test used for normally distributed data and the Wilcoxon signed-rank test for non-normal data. We performed a Chi-square test to analyze the difference in the proportions of neurons responding to visual stimuli between the multisensory discrimination and free-choice groups. Correlation values were computed using Pearson's correlation. All data are presented as mean ± SD for the respective groups. The data were not collected and analyzed in a blinded fashion.

## Acknowledgements

This work was supported by grants from the "STI2030-major projects" (2021ZD0202600), Natural Science Foundation of China (32371046, 31970925, 32271057).

## Additional information

### Funding

| Funder | Grant reference number | Author |
|---|---|---|
| STI2030-major projects | 2021ZD0202600 | Liping Yu |
| Natural Science Foundation of China | 32371046 | Liping Yu |
| Natural Science Foundation of China | 31970925 | Liping Yu |
| Natural Science Foundation of China | 32271057 | Jinghong Xu |

The funders had no role in study design, data collection and interpretation, or the decision to submit the work for publication.

### Author contributions

Song Chang, Resources, Software, Formal analysis, Validation, Visualization, Methodology, Writing – original draft, Project administration; Beilin Zheng, Software, Formal analysis, Validation; Les Keniston, Formal analysis, Writing – original draft, Writing – review and editing; Jinghong Xu, Formal analysis, Funding acquisition, Investigation, Methodology, Writing – original draft, Writing – review and editing; Liping Yu, Conceptualization, Resources, Data curation, Software, Formal analysis, Supervision, Funding acquisition, Validation, Investigation, Visualization, Methodology, Writing – original draft, Project administration, Writing – review and editing

### Author ORCIDs

Jinghong Xu ⓘ https://orcid.org/0000-0002-2864-4196
Liping Yu ⓘ https://orcid.org/0000-0002-9771-5971

### Ethics

The animal procedures conducted in this study were ethically approved by the Local Ethical Review Committee of East China Normal University (Protocol number: m+R20211004) and followed the guidelines outlined in the Guide for the Care and Use of Laboratory Animals of East China Normal University. All surgery was performed under sodium pentobarbital anesthesia, and every effort was made to minimize suffering.

Reviewer #1 (Public review): https://doi.org/10.7554/eLife.102926.3.sa1
Reviewer #2 (Public review): https://doi.org/10.7554/eLife.102926.3.sa2
Reviewer #3 (Public review): https://doi.org/10.7554/eLife.102926.3.sa3
Author response https://doi.org/10.7554/eLife.102926.3.sa4

## Additional files

### Supplementary files

MDAR checklist

Source data 1. Source data contain most of data shown in *Figures 1–7*. However, some exemplar neuron data were not properly saved in the Excel files. These datasets have been deposited separately in Mendeley Data for accessibility.

### Data availability

All experimental data and matlab custom codes have been deposited in Mendeley Data.

The following dataset was generated:

| Author(s) | Year | Dataset title | Dataset URL | Database and Identifier |
|---|---|---|---|---|
| Chang S, Xu J, Yu L | 2025 | Auditory Cortex Learns to Discriminate Audiovisual Cues through Selective Multisensory Enhancement | https://doi.org/10.17632/xm2gfng69b.1 | Mendeley Data, 10.17632/xm2gfng69b.1 |

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
