## [Editor Report · eLife Assessment]

This is an **important** study that aims to investigate the behavioral relevance of multisensory responses recorded in the auditory cortex. The experiments are elegant and well-designed and are supported by appropriate analyses of the data. Although **solid** evidence is presented that is consistent with learning-dependent encoding of visual information in auditory cortex, further work is needed to establish the origin and nature of these non-auditory signals and to definitively rule out any effects of movement-related activity.

---

## [Referee Report · Reviewer #1 (Public review)]

Summary:

Chang and colleagues use tetrode recordings in behaving rats to study how learning an audiovisual discrimination task shapes multisensory interactions in auditory cortex. They find that a significant fraction of neurons in auditory cortex responded to visual (crossmodal) and audiovisual stimuli. Both auditory-responsive and visually-responsive neurons preferentially responded to the cue signaling the contralateral choice in the two-alternative forced choice task. Importantly, multisensory interactions were similarly specific for the congruent audiovisual pairing for the contralateral side.

Strengths:

The experiments are conducted in a rigorous manner. Particularly thorough are the comparisons across cohorts of rats trained in a control task, in a unisensory auditory discrimination task and the multisensory task, while also varying the recording hemisphere and behavioral state (engaged vs. anesthesia). The resulting contrasts strengthen the authors' findings and rule out important alternative explanations regarding the effect of experience. Through the comparisons, they show that the enhancements of activity in multisensory trials in auditory cortex are specific to the paired audiovisual stimulus and specific to contralateral choices in correct trials and thus dependent on learned associations in a task engaged state.

Weaknesses:

The main result that multisensory interactions are specific for contralateral paired audiovisual stimuli is consistent across experiments and interpretable as a learned task-dependent effect. However, the alternative interpretation of behavioral signals is crucial to rule out, which would also be specific to contralateral, correct trials in trained animals. Although the authors focus on the first 150 ms after cue onset, some of the temporal profiles of activity suggest that choice-related activity could confound some of the results.

The main concern (noted by all reviewers) is the interpretation of the evoked activity in visual trials. In the revised manuscript, the authors have not provided much data to disentangle movement related activity from sensory related activity. The only new data is on the visual response dynamics in supplementary figure 2, which is unconvincing both in terms of visual response latency and response dynamics. Therefore, the response of the authors has been insufficient to prove the visual nature of the evoked responses.

In this supplemental figure 2 the same example neuron as in the original manuscript is shown again as well as the average z-scored visual response. First, the visual response latency is inconsistent with literature. The first evoked activity in mouse V1 (!) is routinely reported around 50 ms (for example, 45 ms in Niell Stryker 2008, 52 ms, Schnabel et al. 2018, 54 ms in Oude Lohuis et al. 2024). According to the authors the potential route of crossmodal modulation of AC can occur through either corticocortical connections (which will impose further polysynaptic delays - monosynaptic projection from dLGN or V1 incredibly sparse), or through pulvinar (but pulvinar visual responses are much later (they find 170 vs 80 ms in dLGN, Roth et al. 2019) as expected from a higher order thalamic nucleus). One can also critique the estimation of the response latency which depends on the signal strength (visual response is smaller) and thus choice of threshold. With a different arbitrary threshold one would come to different conclusions.

Second, the temporal response dynamics to visual input are the same as the auditory response. It can be observed that if the data were normalized by the max response the dynamics are very similar, with the response back to near baseline levels at 100 ms post stimulus. I am not aware of publications that have observed response dynamics that are similar between A and V stimuli, nor such short-lasting visual response. In the visual system, mean activity typically drops again around 150-200ms.

With the nature of the observed activity unclear, careful interpretation is warranted about audiovisual interactions in auditory cortex.

---

## [Referee Report · Reviewer #2 (Public review)]

In this revision the authors have made a solid effort to address each of the points raised by all three reviewers. Due to the fact that animals in this study were freely moving, and there has not been any high-speed video recordings to measure whisker movements or other possible stimulus-induced motor effects it is still not possible to rule out motor effects completely. However, the fact that the multisensory enhancements are stimulus specific, much stronger in the multisensory case than the visual only condition, and short in latency it does seem the most parsimonious explanation is likely that these responses are visual in nature.

The delayed auditory stimulus offers some explanation for the very small latency difference between audio and visual stimulus elements. Studies using LED flashes in rat V2 report latencies around ~50 ms (e.g. 2017 paper from Brian Allman's group). The response latencies for visual stimuli in this manuscript are of this order of magnitude, albeit still shorter than that (which presumably means they don't originate from V2).

There are still parts of the manuscript that are inappropriately causal - e.g. line 283 "this suggests that strong multisensory integration is critical for behavior" - it could just as well be the case that high attention / motivation / arousal leads to both strong integration and good behavior.

---

## [Referee Report · Reviewer #3 (Public review)]

Summary:

The manuscript by Chang et al. aims to investigate how the behavioral relevance of auditory and visual stimuli influences the way in which the primary auditory cortex encodes auditory, visual and audiovisual information. The main results is that behavioral training induces an increase in the encoding of auditory and visual information and in multisensory enhancement that is mainly related to the choice located contralaterally with respect to the recorded hemisphere.

Strengths:

The manuscript reports the results of an elegant and well planned experiment meant to investigate if auditory cortex encodes visual information and how learning shapes visual responsiveness in auditory cortex. Analyses are typically well done and properly address the questions raised

Weaknesses:

The authors have addressed most of my comments satisfactorily. However, I am still not convinced by the authors' claim that the use of LED should lead to visually-evoked responses with faster dynamics compared to the use of normal screens. In fact, previous studies using screen-emitted flashed did not report such faster dynamics. Visually-evoked responses in V1 (which are expected to occur earlier than A1) typically do not show onset latencies faster than 40 ms, and have a peak latency of about 100-120 ms. The dynamics shown in the new supplementary Fig. 2 are still faster than this, and thus should be explained. The authors' claim is in fact not supported by cited literature. The authors should at least provide evidence that a similar effect has been observed previously, or otherwise collect evidence themselves. In the absence of such evidence, I remain dubious about the visual nature of the observed activity, especially since, in contrast with what the authors say elsewhere in the rebuttal, involuntary motor reaction to (at least auditory) stimuli can be extremely fast (<40 ms; Clayton et al. 2024) and might thus potentially, at least partially, explain the observed "visual" response.

---

## [Author Response]

The following is the authors’ response to the original reviews

**Reviewer #1 (Public review):**
Summary:Chang and colleagues used tetrode recordings in behaving rats to study how learning an audiovisual discrimination task shapes multisensory interactions in the auditory cortex. They found that a significant fraction of neurons in the auditory cortex responded to visual (crossmodal) and audiovisual stimuli. Both auditory-responsive and visually-responsive neurons preferentially responded to the cue signaling the contralateral choice in the two-alternative forced choice task. Importantly, multisensory interactions were similarly specific for the congruent audiovisual pairing for the contralateral side.Strengths:The experiments were conducted in a rigorous manner. Particularly thorough are the comparisons across cohorts of rats trained in a control task, in a unisensory auditory discrimination task, and the multisensory task, while also varying the recording hemisphere and behavioral state (engaged vs. anesthesia). The resulting contrasts strengthen the authors' findings and rule out important alternative explanations. Through the comparisons, they show that the enhancements of multisensory responses in the auditory cortex are specific to the paired audiovisual stimulus and specific to contralateral choices in correct trials and thus dependent on learned associations in a task-engaged state.

We thank Reviewer #1 for the thorough review and valuable feedback.

Weaknesses:The main result is that multisensory interactions are specific for contralateral paired audiovisual stimuli, which is consistent across experiments and interpretable as a learned task-dependent effect. However, the alternative interpretation of behavioral signals is crucial to rule out, which would also be specific to contralateral, correct trials in trained animals. Although the authors focus on the first 150 ms after cue onset, some of the temporal profiles of activity suggest that choice-related activity could confound some of the results.

We thank the reviewer for raising this important point regarding the potential influence of choice-related activity on our results. In our experimental setup, it is challenging to completely disentangle the effects of behavioral choice from multisensory interaction. However, we conducted relevant analyses to examine the influence of choice-related components on multisensory interaction.

First, we analyzed neural responses during incorrect trials and found a significant reduction in multisensory enhancement for the A^10k^-V^vt^ pairing (Fig. 4). In contrast, for the A^3k^-V^hz^ pairing, there was no strong multisensory interaction during either correct (right direction) or incorrect (left direction) choices. This finding suggests that the observed multisensory interactions are strongly associated with specific cue combinations during correct task performance.

Second, we conducted experiments with unisensory training, in which animals were trained separately on auditory and visual discriminations without explicit multisensory associations. The results demonstrated that unisensory training did not lead to the development of selective multisensory enhancement or congruent auditory-visual preferences, as observed in the multisensory training group. This indicates that the observed multisensory interactions in the auditory cortex are specific to multisensory training and cannot be attributed solely to behavioral signals or choice-related effects.

Finally, we specifically focused on the early 0-150 ms time window after cue onset in our main analyses to minimize contributions from motor-related or decision-related activity, which typically emerge later. This time window allowed us to capture early sensory processing while reducing potential confounds.

Together, these findings strongly suggest that the observed choice-dependent multisensory enhancement is a learned, task-dependent phenomenon that is specific to multisensory training.

The auditory stimuli appear to be encoded by short transient activity (in line with much of what we know about the auditory system), likely with onset latencies (not reported) of 15-30 ms. Stimulus identity can be decoded (Figure 2j) apparently with an onset latency around 50-75 ms (only the difference between A and AV groups is reported) and can be decoded near perfectly for an extended time window, without a dip in decoding performance that is observed in the mean activity Figure 2e. The dynamics of the response of the example neurons presented in Figures 2c and d and the average in 2e therefore do not entirely match the population decoding profile in 2j. Population decoding uses the population activity distribution, rather than the mean, so this is not inherently problematic. It suggests however that the stimulus identity can be decoded from later (choice-related?) activity. The dynamics of the population decoding accuracy are in line with the dynamics one could expect based on choice-related activity. Also the results in Figures S2e,f suggest differences between the two learned stimuli can be in the late phase of the response, not in the early phase.

We appreciate the reviewer’s detailed observations and questions regarding the dynamics of auditory responses and decoding profiles in our study. In our experiment, primary auditory cortex (A1) neurons exhibited short response latencies that meet the established criteria for auditory responses in A1, consistent with findings from many other studies conducted in both anesthetized and task-engaged animals. While the major responses typically occurred during the early period (0-150ms) after cue onset (see population response in Fig. 2e), individual neuronal responses in the whole population were generally dynamic, as illustrated in Figures 2c, 2d, and 3a–c. As the reviewer correctly noted, population decoding leverages the distribution of activity across neurons rather than the mean activity, which explains why the dynamics of population decoding accuracy align well with choice-related activity. This also accounts for the extended decoding window observed in Figure 2j, which does not entirely match the early population response profiles in Figure 2e.

To address the reviewer’s suggestion that differences between the two learned stimuli might arise in the late phase of the response, we conducted a cue selectivity analysis during the 151–300 ms period after cue onset. The results, shown below, indicate that neurons maintained cue selectivity in this late phase for each modality (Supplementary Fig. 5), though the selectivity was lower than in the early phase. However, interpreting this late-phase activity remains challenging. Since A^3k^, V^hz^, and A^3k^-V^hz^ were associated with the right choice, and A^10k^, V^vt^, and A^10k^-V^vt^ with the left choice, it is difficult to disentangle whether the responses reflect choice, sensory features, or a combination of both.

To further investigate, we examined multisensory interactions during the late phase, controlling for choice effects by calculating unisensory and multisensory responses within the same choice context. Our analysis revealed no evident multisensory enhancement for any auditory-visual pairing, nor significant differences between pairings—unlike the robust effects observed in the early phase (Supplementary Fig. 5). We hypothesize that early responses are predominantly sensory-driven and exhibit strong multisensory integration, whereas late responses likely reflect task-related, choice-related, or combined sensory-choice activity, where sensory-driven multisensory enhancement is less prominent. As the focus of this manuscript is on multisensory integration and cue selectivity, we prioritized a detailed analysis of the early phase, where these effects are most prominent. However, the complexity of interpreting late-phase activity remains a challenge and warrants further investigation. We cited Supplementary Fig. 5 in revised manuscript as the following:

“This resulted in a significantly higher mean MSI for the A^10k^-V^vt^ pairing compared to the A^3k^-V^hz^ pairing (0.047 ± 0.124 vs. 0.003 ± 0.096; paired t-test, p < 0.001). Among audiovisual neurons, this biasing is even more pronounced (enhanced vs. inhibited: 62 vs. 2 in A^10k^-V^vt^ pairing, 6 vs. 13 in A^3k^-V^hz^ pairing; mean MSI: 0.119±0.105 in A^10k^-V^vt^ pairing vs. 0.020±0.083 A^3k^-V^hz^ pairing, paired t-test, p<0.00001) (Fig. 3f). Unlike the early period (0-150ms after cue onset), no significant differences in multisensory integration were observed during the late period (151-300ms after cue onset) (Supplementary Fig. 5).”

First, it would help to have the same time axis across panels 2,c,d,e,j,k. Second, a careful temporal dissociation of when the central result of multisensory enhancements occurs in time would discriminate better early sensory processing-related effects versus later decision-related modulations.

Thank you for this valuable feedback. Regarding the first point, we used a shorter time axis in Fig. 2j-k to highlight how the presence of visual cues accelerates the decoding process. This visualization choice was intended to emphasize the early differences in processing speed. For the second point, we have carefully analyzed multisensory integration across different temporal windows. The results presented in the Supplementary Fig. 5 (also see above) already address the late phase, where our data show no evidence of multisensory enhancement for any auditory-visual pairings. This distinction helps clarify that the observed multisensory effects are primarily related to early sensory processing rather than later decision-related modulations. We hope this addresses the concerns raised and appreciate the opportunity to clarify these points.

In the abstract, the authors mention "a unique integration model", "selective multisensory enhancement for specific auditory-visual pairings", and "using this distinct integrative mechanisms". I would strongly recommend that the authors try to phrase their results more concretely, which I believe would benefit many readers, i.e. selective how (which neurons) and specific for which pairings?

We appreciate the reviewer’s suggestion to clarify our phrasing for better accessibility. To address this, we have revised the relevant sentence in the abstract as follows:

"This model employed selective multisensory enhancement for the auditory-visual pairing guiding the contralateral choice, which correlated with improved multisensory discrimination."

**Reviewer #2 (Public review):**
SummaryIn this study, rats were trained to discriminate auditory frequency and visual form/orientation for both unisensory and coherently presented AV stimuli. Recordings were made in the auditory cortex during behaviour and compared to those obtained in various control animals/conditions. The central finding is that AC neurons preferentially represent the contralateral-conditioned stimulus - for the main animal cohort this was a 10k tone and a vertically oriented bar. Over 1/3rd of neurons in AC were either AV/V/A+V and while a variety of multisensory neurons were recorded, the dominant response was excitation by the correctly oriented visual stimulus (interestingly this preference was absent in the visual-only neurons). Animals performing a simple version of the task in which responses were contingent on the presence of a stimulus rather than its identity showed a smaller proportion of AV stimuli and did not exhibit a preference for contralateral conditioned stimuli. The contralateral conditioned dominance was substantially less under anesthesia in the trained animals and was present in a cohort of animals trained with the reverse left/right contingency. Population decoding showed that visual cues did not increase the performance of the decoder but accelerated the rate at which it saturated. Rats trained on auditory and then visual stimuli (rather than simultaneously with A/V/AV) showed many fewer integrative neurons.StrengthsThere is a lot that I like about this paper - the study is well-powered with multiple groups (free choice, reversed contingency, unisensory trained, anesthesia) which provides a lot of strength to their conclusions and there are many interesting details within the paper itself. Surprisingly few studies have attempted to address whether multisensory responses in the unisensory cortex contribute to behaviour - and the main one that attempted to address this question (Lemus et al., 2010, uncited by this study) showed that while present in AC, somatosensory responses did not appear to contribute to perception. The present manuscript suggests otherwise and critically does so in the context of a task in which animals exhibit a multisensory advantage (this was lacking in Lemus et al.,). The behaviour is robust, with AV stimuli eliciting superior performance to either auditory or visual unisensory stimuli (visual were slightly worse than auditory but both were well above chance).

We thank the reviewer for their positive evaluation of our study.

WeaknessesI have a number of points that in my opinion require clarification and I have suggestions for ways in which the paper could be strengthened. In addition to these points, I admit to being slightly baffled by the response latencies; while I am not an expert in the rat, usually in the early sensory cortex auditory responses are significantly faster than visual ones (mirroring the relative first spike latencies of A1 and V1 and the different transduction mechanisms in the cochlea and retina). Yet here, the latencies look identical - if I draw a line down the pdf on the population level responses the peak of the visual and auditory is indistinguishable. This makes me wonder whether these are not sensory responses - yet, they look sensory (very tightly stimulus-locked). Are these latencies a consequence of this being AuD and not A1, or ... ? Have the authors performed movement-triggered analysis to illustrate that these responses are not related to movement out of the central port, or is it possible that both sounds and visual stimuli elicit characteristic whisking movements? Lastly, has the latency of the signals been measured i.e. you generate and play them out synchronously, but is it possible that there is a delay on the audio channel introduced by the amp, which in turn makes it appear as if the neural signals are synchronous? If the latter were the case I wouldn't see it as a problem as many studies use a temporal offset in order to give the best chance of aligning signals in the brain, but this is such an obvious difference from what we would expect in other species that it requires some sort of explanation.

Thank you for your insightful comments. I appreciate the opportunity to clarify these points and strengthen our manuscript. Below, I address your concerns in detail:

We agree that auditory responses are typically faster than visual responses due to the distinct transduction mechanisms. However, in our experiment, we intentionally designed the stimulus setup to elicit auditory and visual responses within a similar time window to maximize the potential for multisensory integration. Specifically, we used pure tone sounds with a 15 ms ramp and visual stimuli generated by an LED array, which produce faster responses compared to mostly used light bars shown on a screen (see Supplementary Fig. 2a). The long ramp of the auditory stimulus slightly delayed auditory response onset, while the LED-generated bar (compared to the bar shown on the screen) elicited visual responses more quickly. This alignment likely facilitated the observed overlap in response latencies.

Neurons’ strong spontaneous activity in freely moving animals complicates the measurement of first spike latencies. Despite that, we still can infer the latency from robust cue-evoked responses. Supplementary Fig. 2b illustrates responses from an exemplar neuron (the same neuron as shown in Fig. 2c), where the auditory response begins 9 ms earlier than the visual response. Given the 28 ms auditory response latency observed here using 15 ms-ramp auditory stimulus, this value is consistent with prior studies in the primary auditory cortex usually using 5 ms ramp pure tones, where latencies typically range from 7 to 28 ms. Across the population (n=559), auditory responses consistently reached 0.5 of the mean Z-scored response 15 ms earlier than visual responses (Supplementary Fig. 2c). The use of Gaussian smoothing in PSTHs supports the reliability of using the 0.5 threshold as an onset latency marker. We cited Supplementary Fig. 2 in the revised manuscript within the Results section (also see the following):

“This suggests multisensory discrimination training enhances visual representation in the auditory cortex. To optimize the alignment of auditory and visual responses and reveal the greatest potential for multisensory integration, we used long-ramp pure tone auditory stimuli and quick LED-array-elicited visual stimuli (Supplementary Fig. 2). While auditory responses were still slightly earlier than visual responses, the temporal alignment was sufficient to support robust integration.”

We measured the time at which rats left the central port and confirmed that these times occur significantly later than the neuronal responses analyzed (see Fig. 1c-d). While we acknowledge the potential influence of movements such as whiskering, facial movements, head direction changes, or body movements on neuronal responses, precise monitoring of these behaviors in freely moving animals remains a technical challenge. However, given the tightly stimulus-locked nature of the neuronal responses observed, we believe they primarily reflect sensory processing rather than movement-related activity.

To ensure accurate synchronization of auditory and visual stimuli, we verified the latencies of our signals. The auditory and visual stimuli were generated and played out synchronously with no intentional delay introduced. The auditory amplifier used in our setup introduces minimal latency, and any such delay would have been accounted for during calibration. Importantly, even if a small delay existed, it would not undermine our findings, as many studies intentionally use temporal offsets to facilitate alignment of neural signals. Nonetheless, the temporal overlap observed here is primarily a result of our experimental design aimed at promoting multisensory integration.

We hope these clarifications address your concerns and highlight the robustness of our findings.

Reaction times were faster in the AV condition - it would be of interest to know whether this acceleration is sufficient to violate a race model, given the arbitrary pairing of these stimuli. This would give some insight into whether the animals are really integrating the sensory information. It would also be good to clarify whether the reaction time is the time taken to leave the center port or respond at the peripheral one.

We appreciate your request for clarification. In our analysis, reaction time (RT) is defined as the time taken for the animal to leave the center port after cue onset. This measure was chosen because it reflects the initial decision-making process and the integration of sensory information leading to action initiation. The time taken to respond at the peripheral port, commonly referred to as movement time, was not included in our RT measure. However, movement time data is available in our dataset, and we are open to further analysis if deemed necessary.

To determine whether the observed acceleration in RTs in the audiovisual (AV) condition reflects true multisensory integration rather than statistical facilitation, we tested for violations of the race model inequality (Miller, 1982). This approach establishes a bound for the probability of a response occurring within a given time interval under the assumption that the auditory (A) and visual (V) modalities operate independently. Specifically, we calculated cumulative distribution functions (CDFs) for the RTs in the A, V, and AV conditions (please see Author response image 1). In some rats, the AV_RTs exceeded the race model prediction at multiple time points, suggesting that the observed acceleration is not merely due to statistical facilitation but reflects true multisensory integration. Examples of these violations are shown in Panels a-b of the following figure. However, in other rats, the AV_RTs did not exceed the race model prediction, as illustrated in Author response image 1c-d.

This variability may be attributed to task-specific factors in our experimental design. For instance, the rats were not under time pressure to respond immediately after cue onset, as the task emphasized accuracy over speed. This lack of urgency may have influenced their behavioral responses and movement patterns. The race model is typically applied to assess multisensory integration in tasks where rapid responses are critical, often under conditions that incentivize speed (e.g., time-restricted tasks). In our study, the absence of strict temporal constraints may have reduced the likelihood of observing consistent violations of the race model. Furthermore, In our multisensory discrimination task, animals should discriminate multiple cues and make a behavioral choice have introduced additional variability in the degree of integration observed across individual animals. Additionally, factors such as a decline in thirst levels and physical performance as the task progressed may have significantly contributed to the variability in our results. These considerations are important for contextualizing the race model findings and interpreting the data within the framework of our experimental design.

**Author response image 1. sa4fig1:** Reaction time cumulative distribution functions (CDFs) and race model evaluation. (a) CDFs of reaction times (RTs) for auditory (blue), visual (green), and audiovisual stimuli (red) during the multisensory discrimination task. The summed CDF of the auditory and visual conditions (dashed purple, CDF_Miller) represents the race model prediction under independent sensory processing. The dashed yellow line represents the CDF of reaction times predicted by the race model. According to the race model inequality, the CDF for audiovisual stimuli (CDF_AV) should always lie below or to the right of the sum of CDF_A and CDF_V. In this example, the inequality is violated at nearly t = 200 ms, where CDF_AV is above CDF_Miller. (b) Data from another animal, showing similar results. (c, d) CDFs of reaction times for two other animals. In these cases, the CDFs follow the race model inequality, with CDF_AV consistently lying below or to the right of CDF_A + CDF_V.

The manuscript is very vague about the origin or responses - are these in AuD, A1, AuV... ? Some attempts to separate out responses if possible by laminar depth and certainly by field are necessary. It is known from other species that multisensory responses are more numerous, and show greater behavioural modulation in non-primary areas (e.g. Atilgan et al., 2018).

Thank you for highlighting the importance of specifying the origin of the recorded responses. In the manuscript, we have detailed the implantation process in both the Methods and Results sections, indicating that the tetrode array was targeted to the primary auditory cortex. Using a micromanipulator (RWD, Shenzhen, China), the tetrode array was precisely positioned at stereotaxic coordinates 3.5–5.5 mm posterior to bregma and 6.4 mm lateral to the midline, and advanced to a depth of approximately 2–2.8 mm from the brain surface, corresponding to the primary auditory cortex. Although our recordings were aimed at A1, it is likely that some neurons from AuD and/or AuV were also included due to the anatomical proximity.

In fact, in our unpublished data collected from AuD, we observed that over 50% of neurons responded to or were modulated by visual cues, consistent with findings from many other studies. This suggests that visual representations are more pronounced in AuD compared to A1. However, as noted in the manuscript, our primary focus was on A1, where we observed relatively fewer visual or audiovisual modulations in untrained rats.

Regarding laminar depth, we regret that we were unable to determine the specific laminar layers of the recorded neurons in this study, a limitation primarily due to the constraints of our recording setup.

**Reviewer #3 (Public review):**
Summary:The manuscript by Chang et al. aims to investigate how the behavioral relevance of auditory and visual stimuli influences the way in which the primary auditory cortex encodes auditory, visual, and audiovisual information. The main result is that behavioral training induces an increase in the encoding of auditory and visual information and in multisensory enhancement that is mainly related to the choice located contralaterally with respect to the recorded hemisphere.Strengths:The manuscript reports the results of an elegant and well-planned experiment meant to investigate if the auditory cortex encodes visual information and how learning shapes visual responsiveness in the auditory cortex. Analyses are typically well done and properly address the questions raised.

We sincerely thank the reviewer for their thoughtful and positive evaluation of our study.

Weaknesses:Major(1) The authors apparently primarily focus their analyses of sensory-evoked responses in approximately the first 100 ms following stimulus onset. Even if I could not find an indication of which precise temporal range the authors used for analysis in the manuscript, this is the range where sensory-evoked responses are shown to occur in the manuscript figures. While this is a reasonable range for auditory evoked responses, the same cannot be said for visual responses, which commonly peak around 100-120 ms, in V1. In fact, the latency and overall shape of visual responses are quite different from typical visual responses, that are commonly shown to display a delay of up to 100 ms with respect to auditory responses. All traces that the authors show, instead, display visual responses strikingly overlapping with auditory ones, which is not in line with what one would expect based on our physiological understanding of cortical visually-evoked responses. Similarly, the fact that the onset of decoding accuracy (Figure 2j) anticipates during multisensory compared to auditory-only trials is hard to reconcile with the fact that visual responses have a later onset latency compared to auditory ones. The authors thus need to provide unequivocal evidence that the results they observe are truly visual in origin. This is especially important in view of the ever-growing literature showing that sensory cortices encode signals representing spontaneous motor actions, but also other forms of non-sensory information that can be taken prima facie to be of sensory origin. This is a problem that only now we realize has affected a lot of early literature, especially - but not only - in the field of multisensory processing. It is thus imperative that the authors provide evidence supporting the true visual nature of the activity reported during auditory and multisensory conditions, in both trained, free-choice, and anesthetized conditions. This could for example be achieved causally (e.g. via optogenetics) to provide the strongest evidence about the visual nature of the reported results, but it's up to the authors to identify a viable solution. This also applies to the enhancement of matched stimuli, that could potentially be explained in terms of spontaneous motor activity and/or pre-motor influences. In the absence of this evidence, I would discourage the author from drawing any conclusion about the visual nature of the observed activity in the auditory cortex.

We thank the reviewers for highlighting the critical issue of validating the sensory origin of the reported responses, particularly regarding the timing of visual responses and the potential confound of motor-related activity.

We analyzed neural responses within the first 150 ms following cue onset, as stated in the manuscript. This temporal window encompasses the peak of visual responses. The responses to visual stimuli occur predominantly within the first 100 ms after cue onset, preceding the initiation of body movements in behavioral tasks. This temporal dissociation aligns with previous studies, which demonstrate that motor-related activity in sensory cortices generally emerges later and is often associated with auditory rather than visual stimuli

We acknowledge that auditory responses are typically faster than visual responses due to distinct transduction mechanisms. However, in our experiment, we intentionally designed the stimulus setup to elicit auditory and visual responses within a similar time window to maximize the potential for multisensory integration. Specifically, we used pure tone sounds with a 15 ms ramp and visual stimuli generated by an LED array, which produce faster responses compared to commonly used light bars shown on a screen. The long ramp of the auditory stimulus slightly delayed auditory response onset, while the LED-generated bar elicited visual responses more quickly (Supplementary Fig. 2). This alignment facilitated the observed overlap in response latencies. As we measured in neurons with robust visual response, first spike latencies is approximately 40 ms, as exemplified by a neuron with a low spontaneous firing rate and a strong, stimulus-evoked response (Supplementary Fig. 4). Across the population (n = 559 neurons), auditory responses reached 0.5 of the mean Z-scored response 15 ms earlier than visual responses on average (Supplementary Fig. 2). We cited Supplementary Fig. 4 in the Results section as follows:

“Regarding the visual modality, 41% (80/196) of visually-responsive neurons showed a significant visual preference (Fig. 2f). The visual responses observed within the 0–150 ms window after cue onset were consistent and unlikely to result from visually evoked movement-related activity. This conclusion is supported by the early timing of the response (Fig. 2e) and exemplified by a neuron with a low spontaneous firing rate and a robust, stimulus-evoked response (Supplementary Fig. 4).”

We acknowledge the growing body of literature suggesting that sensory cortices can encode signals related to motor actions or non-sensory factors. To address this concern, we emphasize that visual responses were present not only during behavioral tasks but also in anesthetized conditions, where motor-related signals are absent. Additionally, movement-evoked responses tend to be stereotyped and non-discriminative. In contrast, the visual responses observed in our study were highly consistent and selective to visual cue properties, further supporting their sensory origin.

In summary, the combination of anesthetized and behavioral recordings, the temporal profile of responses, and their discriminative nature strongly support the sensory (visual) origin of the observed activity within the early response period. While the current study provides strong temporal and experimental evidence for the sensory origin of the visual responses, we agree that causal approaches, such as optogenetic silencing of visual input, could provide even stronger validation. Future work will explore these methods to further dissect the visual contributions to auditory cortical activity.

(2) The finding that AC neurons in trained mice preferentially respond - and enhance - auditory and visual responses pertaining to the contralateral choice is interesting, but the study does not show evidence for the functional relevance of this phenomenon. As has become more and more evident over the past few years (see e.g. the literature on mouse PPC), correlated neural activity is not an indication of functional role. Therefore, in the absence of causal evidence, the functional role of the reported AC correlates should not be overstated by the authors. My opinion is that, starting from the title, the authors need to much more carefully discuss the implications of their findings.

We fully agree that correlational data alone cannot establish causality. In light of your suggestion, we will revise the manuscript to more carefully discuss the implications of our findings, acknowledging that the preferred responses observed in AC neurons, particularly in relation to the contralateral choice, are correlational. We have updated several sentences in the manuscript to avoid overstating the functional relevance of these observations. Below are the revisions we have made:

Abstract section

"Importantly, many audiovisual neurons in the AC exhibited experience-dependent associations between their visual and auditory preferences, displaying a unique integration model. This model employed selective multisensory enhancement for the auditory-visual pairing guiding the contralateral choice, which correlated with improved multisensory discrimination."

(Page 8, fourth paragraph in Results Section)

"This aligns with findings that neurons in the AC and medial prefrontal cortex selectively preferred the tone associated with the behavioral choice contralateral to the recorded cortices during sound discrimination tasks, potentially reflecting the formation of sound-to-action associations. However, this preference represents a neural correlate, and further work is required to establish its causal link to behavioral choices."

(rewrite 3rd paragraph in Discussion Section)

"Consistent with prior research(10,31), most AC neurons exhibited a selective preference for cues associated with contralateral choices, regardless of the sensory modality. This suggests that AC neurons may contribute to linking sensory inputs with decision-making, although their causal role remains to be examined. "

"These results indicate that multisensory training could drive the formation of specialized neural circuits within the auditory cortex, facilitating integrated processing of related auditory and visual information. However, further causal studies are required to confirm this hypothesis and to determine whether the auditory cortex is the primary site of these circuit modifications."

MINOR:(1) The manuscript is lacking what pertains to the revised interpretation of most studies about audiovisual interactions in primary sensory cortices following the recent studies revealing that most of what was considered to be crossmodal actually reflects motor aspects. In particular, recent evidence suggests that sensory-induced spontaneous motor responses may have a surprisingly fast latency (within 40 ms; Clayton et al. 2024). Such responses might also underlie the contralaterally-tuned responses observed by the authors if one assumes that mice learn a stereotypical response that is primed by the upcoming goal-directed, learned response. Given that a full exploration of this issue would require high-speed tracking of orofacial and body motions, the authors should at least revise the discussion and the possible interpretation of their results not just on the basis of the literature, but after carefully revising the literature in view of the most recent findings, that challenge earlier interpretations of experimental results.

Thank you for pointing out this important consideration. We have revised the discussion (paragraph 8-9) as follows:

“There is ongoing debate about whether cross-sensory responses in sensory cortices predominantly reflect sensory inputs or are influenced by behavioral factors, such as cue-induced body movements. A recent study shows that sound-clip evoked activity in visual cortex have a behavioral rather than sensory origin and is related to stereotyped movements(48). Several studies have demonstrated sensory neurons can encode signals associated with whisking(49), running(50), pupil dilation 510 and other movements(52). In our study, the responses to visual stimuli in the auditory cortex occurred primarily within a 100 ms window following cue onset. This early timing suggests that the observed responses likely reflect direct sensory inputs, rather than being modulated by visually-evoked body or orofacial movements, which typically occur with a delay relative to sensory cue onset(53).

A recent study by Clayton et al. (2024) demonstrated that sensory stimuli can evoke rapid motor responses, such as facial twitches, within 50 ms, mediated by subcortical pathways and modulated by descending corticofugal input(56). These motor responses provide a sensitive behavioral index of auditory processing. Although Clayton et al. did not observe visually evoked facial movements, it is plausible that visually driven motor activity occurs more frequently in freely moving animals compared to head-fixed conditions. In goal-directed tasks, such rapid motor responses might contribute to the contralaterally tuned responses observed in our study, potentially reflecting preparatory motor behaviors associated with learned responses. Consequently, some of the audiovisual integration observed in the auditory cortex may represent a combination of multisensory processing and preparatory motor activity. Comprehensive investigation of these motor influences would require high-speed tracking of orofacial and body movements. Therefore, our findings should be interpreted with this consideration in mind. Future studies should aim to systematically monitor and control eye, orofacial, and body movements to disentangle sensory-driven responses from motor-related contributions, enhancing our understanding of motor planning’s role in multisensory integration.”

(2) The methods section is a bit lacking in details. For instance, information about the temporal window of analysis for sensory-evoked responses is lacking. Another example: for the spike sorting procedure, limited details are given about inclusion/exclusion criteria. This makes it hard to navigate the manuscript and fully understand the experimental paradigm. I would recommend critically revising and expanding the methods section.

Thank you for raising this point. We clarified the temporal window by including additional details in the methods section, even though this information was already mentioned in the results section. Specifically, we now state:

(Neural recordings and Analysis in methods section)

“...These neural signals, along with trace signals representing the stimuli and session performance information, were transmitted to a PC for online observation and data storage. Neural responses were analyzed within a 0-150ms temporal window after cue onset, as this period was identified as containing the main cue-evoked responses for most neurons. This time window was selected based on the consistent and robust neural activity observed during this period.”

We appreciate your concern regarding spike sorting procedure. To address this, we have expanded the methods section to provide more detailed information about the quality of our single-unit recordings. we have added detailed information in the text, as shown below (Analysis of electrophysiological data in methods section):

“Initially, the recorded raw neural signals were band-pass filtered in the range of 300-6000 Hz to eliminate field potentials. A threshold criterion, set at no less than three times the standard deviation (SD) above the background noise, was applied to automatically identify spike peaks. The detected spike waveforms were then subjected to clustering using template-matching and built-in principal component analysis tool in a three-dimensional feature space. Manual curation was conducted to refine the sorting process. Each putative single unit was evaluated based on its waveform and firing patterns over time. Waveforms with inter-spike intervals of less than 2.0 ms were excluded from further analysis. Spike trains corresponding to an individual unit were aligned to the onset of the stimulus and grouped based on different cue and choice conditions. Units were included in further analysis only if their presence was stable throughout the session, and their mean firing rate exceeded 2 Hz. The reliability of auditory and visual responses for each unit was assessed, with well-isolated units typically showing the highest response reliability.”

**Reviewer #1 (Recommendations for the authors):**
(1) Some of the ordering of content in the introduction could be improved. E.g. line 49 reflects statements about the importance of sensory experience, which is the topic of the subsequent paragraph. In the discussion, line 436, there is a discussion of the same findings as line 442. These two paragraphs in general appear to discuss similar content. Similarly, the paragraph starting at line 424 and at line 451 both discuss the plasticity of multisensory responses through audiovisual experience, as well as the paragraph starting at line 475 (but now audiovisual pairing is dubbed semantic). In the discussion of how congruency/experience shapes multisensory interactions, the authors should relate their findings to those of Meijer et al. 2017 and Garner and Keller 2022 (visual cortex) about enhanced and suppressed responses and their potential role (as well as other literature such as Banks et al. 2011 in AC).

We thank the reviewer for their detailed observations and valuable recommendations to improve the manuscript's organization. Below, we address each point:

We deleted the sentence, "Sensory experience has been shown to shape cross-modal presentations in sensory cortices" (Line 49), as the subsequent paragraph discusses sensory experience in detail.

To avoid repetition, we removed the sentence, "This suggests that multisensory training enhances AC's ability to process visual information" (Lines 442–443).

Regarding the paragraph starting at Line 475, we believe its current form is appropriate, as it focuses on the influence of semantic congruence on multisensory integration, which differs from the topics discussed in the other paragraphs.

We have cited the three papers suggested by the reviewer in the appropriate sections of the manuscript.

(Paragraph 6 in discussion section)

“…A study conducted on the gustatory cortex of alert rats has shown that cross-modal associative learning was linked to a dramatic increase in the prevalence of neurons responding to nongustatory stimuli (24). Moreover, in the primary visual cortex, experience-dependent interactions can arise from learned sequential associations between auditory and visual stimuli, mediated by corticocortical connections rather than simultaneous audiovisual presentations (26).”

(Paragraph 2 in discussion section)

“...Meijer et al. reported that congruent audiovisual stimuli evoke balanced enhancement and suppression in V1, while incongruent stimuli predominantly lead to suppression(6), mirroring our findings in AC, where multisensory integration was dependent on stimulus feature…”

(Paragraph 2 in introduction section)

“...Anatomical investigations reveal reciprocal nerve projections between auditory and visual cortices(4,11-15), highlighting the interconnected nature of these sensory systems. Moreover, two-photon calcium imaging in awake mice has shown that audiovisual encoding in the primary visual cortex depends on the temporal congruency of stimuli, with temporally congruent audiovisual stimuli eliciting balanced enhancement and suppression, whereas incongruent stimuli predominantly result in suppression(6).”

(2) The finding of purely visually responsive neurons in the auditory cortex that moreover discriminate the stimuli is surprising given previous results (Iurilli et al. 2012, Morrill and Hasenstaub 2018 (only L6), Oude Lohuis et al. 2024, Atilgan et al. 2018, Chou et al. 2020). Reporting the latency of this response is interesting information about the potential pathways by which this information could reach the auditory system. Furthermore, spike isolation quality and histological verification are described in little detail. It is crucial for statements about the auditory, visual, or audiovisual response of individual neurons to substantiate the confidence level about the quality of single-unit recordings and where they were recorded. Do the authors have data to support that visual and audiovisual responses were not restricted to posteromedial tetrodes or clusters with poor quality? A discussion of finding V-responsive units in AC with respect to literature is warranted. Furthermore, the finding that also in visual trials behaviorally relevant information about the visual cue (with a bias for the contralateral choice cue) is sent to the AC is pivotal in the interpretation of the results, which as far as I note not really considered that much.

We appreciate the reviewer’s thoughtful comments and have addressed them as follows:

Discussion of finding choice-related V-responsive units in AC with respect to literature and potential pathways

3rd paragraph in the Discussion section

“Consistent with prior research(10,31), most AC neurons exhibited a selective preference for cues associated with contralateral choices, regardless of the sensory modality. This suggests that AC neurons may contribute to linking sensory inputs with decision-making, although their causal role remains to be examined. Associative learning may drive the formation of new connections between sensory and motor areas of the brain, such as cortico-cortical pathways(35). Notably, this cue-preference biasing was absent in the free-choice group. A similar bias was also reported in a previous study, where auditory discrimination learning selectively potentiated corticostriatal synapses from neurons representing either high or low frequencies associated with contralateral choices(32)…”

6th paragraph in the Discussion section

“Our results extend prior finding(4,47), showing that visual input not only reaches the AC but can also drive discriminative responses, particularly during task engagement. This task-specific plasticity enhances cross-modal integration, as demonstrated in other sensory systems. For example, calcium imaging studies in mice showed that a subset of multimodal neurons in visual cortex develops enhanced auditory responses to the paired auditory stimulus following coincident auditory–visual experience(25)…”

8th paragraph in the Discussion section

“…In our study, the responses to visual stimuli in the auditory cortex occurred primarily within a 100 ms window following cue onset, suggesting that visual information reaches the AC through rapid pathways. Potential candidates include direct or fast cross-modal inputs, such as pulvinar-mediated pathways(8) or corticocortical connections(5，54), rather than slower associative mechanisms. This early timing indicates that the observed responses were less likely modulated by visually-evoked body or orofacial movements, which typically occur with a delay relative to sensory cue onset(55).”

Response Latency

Regarding the latency of visually driven responses, we have included this information in our response to the second reviewer’s first weakness (please see the above). Briefly, we analyzed neural responses within a 0-150ms temporal window after cue onset, as this period captures the most consistent and robust cue-evoked responses across neurons.

Purely Visually Responsive Neurons in A1

We agree that the finding of visually responsive neurons in the auditory cortex may initially seem surprising. However, these neurons might not have been sensitive to target auditory cues in our task but could still respond to other sound types. Cortical neurons are known to exhibit significant plasticity during the cue discrimination tasks, as well as during passive sensory exposure. Thus, the presence of visually responsive neurons is not inconsistent with prior findings but highlights task-specific sensory tuning. We confirm that responses were not restricted to posteromedial tetrodes or low-quality clusters (see an example of a robust visually responsive neuron in supplementary Fig. 4). Histological analysis verified electrode placements across the auditory cortex.

For spike sorting, we have added detailed information in the text, as shown below:

“Initially, the recorded raw neural signals were band-pass filtered in the range of 300-6000 Hz to eliminate field potentials. A threshold criterion, set at no less than three times the standard deviation (SD) above the background noise, was applied to automatically identify spike peaks. The detected spike waveforms were then subjected to clustering using template-matching and built-in principal component analysis tool in a three-dimensional feature space. Manual curation was conducted to refine the sorting process. Each putative single unit was evaluated based on its waveform and firing patterns over time. Waveforms with inter-spike intervals of less than 2.0 ms were excluded from further analysis. Spike trains corresponding to an individual unit were aligned to the onset of the stimulus and grouped based on different cue and choice conditions. Units were included in further analysis only if their presence was stable throughout the session, and their mean firing rate exceeded 2 Hz. The reliability of auditory and visual responses for each unit was assessed, with well-isolated units typically showing the highest response reliability.”

(3) In the abstract it seems that in "Additionally, AC neurons..." the connective word 'additionally' is misleading as it is mainly a rephrasing of the previous statement.

Replaced "Additionally" with "Furthermore" to better signal elaboration and continuity.

(4) The experiments included multisensory conflict trials - incongruent audiovisual stimuli. What was the behavior for these trials given multiple interesting studies on the neural correlates of sensory dominance (Song et al. 2017, Coen et al. 2023, Oude Lohuis et al. 2024).

We appreciate your feedback and have addressed it by including a new figure (supplemental Fig. 8) that illustrates choice selection during incongruent audiovisual stimuli. Panel (a) shows that rats displayed confusion when exposed to mismatched stimuli, resulting in choice patterns that differed from those observed in panel (b), where consistent audiovisual stimuli were presented. To provide clarity and integrate this new figure effectively into the manuscript, we updated the results section as follows:

“...Rats received water rewards with a 50% chance in either port when an unmatched multisensory cue was triggered. Behavioral analysis revealed that Rats displayed notable confusion in response to unmatched multisensory cues, as evidenced by their inconsistent choice patterns (supplementary Fig. 8).”

(5) Line 47: The AC does not 'perceive' sound frequency, individual brain regions are not thought to perceive.

e appreciate the reviewer’s observation and have revised the sentence to ensure scientific accuracy. The updated sentence in the second paragraph of the Introduction now reads:

“Even irrelevant visual cues can affect sound discrimination in AC^10^.”

(6) Line 59-63: The three questions are not completely clear to me. Both what they mean exactly and how they are different. E.g. Line 60: without specification, it is hard to understand which 'strategies' are meant by the "same or different strategies"? And Line 61: What is meant by the quotation marks for match and mismatch? I assume this is referring to learned congruency and incongruency, which appears almost the same question as number 3 (how learning affects the cortical representation).

We have revised the three questions for improved clarity and distinction as follows:

“This limits our understanding of multisensory integration in sensory cortices, particularly regarding: (1) Do neurons in sensory cortices adopt consistent integration strategies across different audiovisual pairings, or do these strategies vary depending on the pairing? (2) How does multisensory perceptual learning reshape cortical representations of audiovisual objects? (3) How does the congruence between auditory and visual features—whether they "match" or "mismatch" based on learned associations—impact neural integration?”

(7) Is the data in Figures 1c and d only hits?

Only correct trials are included. We add this information in the figure legend. Please see Fig. 1 legend. Also, please see below

“c Cumulative frequency distribution of reaction time (time from cue onset to leaving the central port) for one representative rat in auditory, visual and multisensory trials (correct only). d Comparison of average reaction times across rats in auditory, visual, and multisensory trials (correct only).”

(8) Figure S1b: Preferred frequency is binned in non-equidistant bins, neither linear nor logarithmic. It is unclear what the reason is.

The edges of the bins for the preferred frequency were determined based on a 0.5-octave increment, starting from the smallest boundary of 8 kHz. Specifically, the bin edges were calculated as follows:

8×2^0.5^=11.3 kHz;

8×2^1^=16 kHz;

8×2^1.5^=22.6 kHz;

8×2^2^=32 kHz;

This approach reflects the common practice of using changes in octaves to define differences between pure tone frequencies, as it aligns with the logarithmic perception of sound frequency in auditory neuroscience.

(9) Figure S1d: why are the responses all most neurons very strongly correlated given the frequency tuning of A1 neurons? Further, the mean normalized response presented in Figure S2e does seem to indicate a stronger response for 10kHz tones than 3kHz, in conflict with the data from anesthetized rats presented in Figure S2e.

There is no discrepancy in the data. In Figure S1d, we compared neuronal responses to 10 kHz and 3 kHz tones, demonstrating that most neurons responded well to both frequencies. This panel does not aim to illustrate frequency selectivity but rather the overall responsiveness of neurons to these tones. For detailed information on sound selectivity, readers can refer to Figures S3a-b, which show that while more neurons preferred 10 kHz tones, the proportion is lower than in neurons recorded during the multisensory discrimination task. This distinction explains the observed differences and aligns with the results presented.

(10) Line 79: For clarity, it can be added that the multisensory trials presented are congruent trials (jointly indicated rewarded port), and perhaps that incongruent trials are discussed later in the paper.

We believe additional clarification is unnecessary, as the designations "A^3k^V^hz^" and "A^10k^V^vt^" clearly indicate the specific combinations of auditory and visual cues presented during congruent trials. Additionally, the discussion of incongruent trials is provided later in the manuscript, as noted by the reviewer.

(11) Line 111: the description leaves unclear that the 35% reflects the combination of units responsive to visual only and responsive to auditory or visual.

The information is clearly presented in Figure 2b, which shows the proportions of neurons responding to auditory-only (A), visual-only (V), both auditory and visual (A, V), and audiovisual-only (VA) stimuli in a pie chart. Readers can refer to this figure for a detailed breakdown of the neuronal response categories.

(12) Figure 2h: consider a colormap with diverging palette and equal positive and negative maximum (e.g. -0.6 to 0.6) and perhaps reiterate in the color bar legend which stimulus is preferred for which selectivity index.

We appreciate the suggestion; however, we believe that the current colormap effectively conveys the data and the intended interpretation. The existing color bar legend already provides clear information about the selectivity index, and the stimulus preference is adequately explained in the figure caption. As such, further adjustments are not necessary.

(13) Line 160: "a ratio of 60:20 for V^vt^ 160 preferred vs. V^hz^ preferred neurons." Is this supposed to add up to 100, or is this a ratio of 3:1?

We rewrite the sentence. Please see below:

“Similar to the auditory selectivity observed, a greater proportion of neurons favored the visual stimulus (V^vt^) associated with the contralateral choice, with a 3:1 ratio of V^vt^-preferred to V^hz^-preferred neurons.”

(14) The statement in Figure 2g and line 166/167 could be supported by a statistical test (chi-square?).

Thank you for the suggestion. However, we believe that a statistical test is not required in this case, as the patterns observed are clearly represented in Figure 2g. The qualitative differences between the groups are evident and sufficiently supported by the data.

(15) Line 168, it is unclear in what sense 'dominant' is meant. Is audition perceived as a dominant sensory modality in a behavioral sense (e.g. Song et al. 2017), or are auditory signals the dominant sensory signal locally in the auditory cortex?

Thank you for the clarification. To address your question, by "dominant," we are referring to the fact that auditory inputs are the most prominent and influential among the sensory signals feeding into the auditory cortex. This reflects the local dominance of auditory signals within the auditory cortex, rather than a behavioral dominance of auditory perception. We have revised the sentence as follows:

“We propose that the auditory input, which dominates within the auditory cortex, acts as a 'teaching signal' that shapes visual processing through the selective reinforcement of specific visual pathways during associative learning.”

(16) Line 180: "we discriminated between auditory, visual, and multisensory cues." This phrasing indicated that the SVMs were trained to discriminate sensory modalities (as is done later in the manuscript), rather than what was done: discriminate stimuli within different categories of trials.

Thank you for your comment. We have revised the sentence for clarity. Please see the updated version below:

“Using cross-validated support vector machine (SVM) classifiers, we examined how this pseudo-population discriminates stimulus identity within the same modality (e.g., A^3k^ vs. A^10k^ for auditory stimuli, V^hz^ vs. V^vt^ for visual stimuli, A^3k^V^hz^ vs. A^10k^V^vt^ for multisensory stimuli).”

(17) Line 185: "a deeply accurate incorporation of visual processing in the auditory cortex." the phrasing is a bit excessive for a binary classification performance.

Thank you for pointing this out. We have revised the sentence to better reflect the findings without overstating them:

“Interestingly, AC neurons could discriminate between two visual targets with around 80% accuracy (Fig. 2j), demonstrating a meaningful incorporation of visual information into auditory cortical processing.”

(18) Figure 3, title. An article is missing (a,an/the).

Done. Please see below:

**“**Fig. 3 Auditory and visual integration in the multisensory discrimination task**”**

(19) Line 209, typo pvalue: p<-0.00001.

Done (p<0.00001).

(20) Line 209, the pattern is not weaker. The pattern is the same, but more weakly expressed.

Thank you for your valuable feedback. We appreciate your clarification and agree that our phrasing could be improved for accuracy. The observed pattern under anesthesia is indeed the same but less strongly expressed compared to the task engagement. We have revised the sentence to better reflect this distinction:

“A similar pattern, albeit less strongly expressed, was observed under anesthesia (Supplementary Fig. 3c-3f), suggesting that multisensory perceptual learning may induce plastic changes in AC.”

(21) Line 211: choice-free group → free-choice group.

Done.

(22) Line 261: wrong → incorrect (to maintain consistent terminology).

Done.

(23) Line 265: why 'likely'? Are incorrect choices on the A^3k^-V^hz^ trials not by definition contralateral and vice versa? Or are there other ways to have incorrect trials?

We deleted the word of ‘likely’. Please see below:

“…, correct choices here correspond to ipsilateral behavioral selection, while incorrect choices correspond to contralateral behavioral selection.”

(24) Typo legend Fig 3a-c (tasks → task). (only one task performed).

Done.

(25) Line 400: typo: Like → like.

Done.

(26) Line 405: What is meant by a cohesive visual stimulus? Congruent? Rephrase.

Done. Please see the below:

“…layer 2/3 neurons of the primary visual cortex(7), and a **congruent** visual stimulus can enhance sound representation…”

(27) Line 412: Very general statement and obviously true: depending on the task, different sensory elements need to be combined to guide adaptive behavior.

We really appreciate the reviewer and used this sentence (see second paragraph in discussion section).

(28) Line 428: within → between (?).

Done.

(29) Figure 3L is not referenced in the main text. By going through the figures and legends my understanding is that this shows that most neurons have a multisensory response that lies between 2 z-scores of the predicted response in the case of 83% of the sum of the auditory and the visual response. However, how was the 0.83 found? Empirically? Figure S3 shows a neuron that does follow a 100% summation. Perhaps the authors could quantitatively support their estimate of 83% of the A + V sum, by varying the fraction of the sum (80%, 90%, 100% etc.) and showing the distribution of the preferred fraction of the sum across neurons, or by showing the percentage of neurons that fall within 2 z-scores for each of the fractions of the sum.

Thank you for your detailed feedback and suggestions regarding Figure 3L and the 83% multiplier.

(1) Referencing Figure 3L:

Figure 3L is referenced in the text. To enhance clarity, we have revised the text to explicitly highlight its relevance:

“Specifically, as illustrated in Fig. 3k, the observed multisensory response approximated 83% of the sum of the auditory and visual responses in most cases, as quantified in Fig. 3L.”

(2) Determination of the 0.83 Multiplier:

The 0.83 multiplier was determined empirically by comparing observed audiovisual responses with the predicted additive responses (i.e., the sum of auditory and visual responses). For each neuron, we calculated the auditory, visual, and audiovisual responses. We then compared the observed audiovisual response with scaled sums of auditory and visual responses (Fig. 3k), expressed as fractions of the additive prediction (e.g., 0.8, 0.83, 0.9, etc.). We found that when the scaling factor was 0.83, the population-wide difference between predicted and observed multisensory responses, expressed as z-scores, was minimized. Specifically, at this value, the mean z-score across the population was approximately zero (-0.0001±1.617), indicating the smallest deviation between predicted and observed responses.

(30) Figure 5e: how come the diagonal has 0.5 decoding accuracy within a category? Shouldn't this be high within-category accuracy? If these conditions were untested and it is an issue of the image display it would be informative to test the cross-validated performance within the category as well as a benchmark to compare the across-category performance to. Aside, it is unclear which conventions from Figure 2 are meant by the statement that conventions were the same.

The diagonal values (~0.5 decoding accuracy) within each category reflect chance-level performance. This occurs because the decoder was trained and tested on the same category conditions in a cross-validated manner, and within-category stimulus discrimination was not the primary focus of our analysis. Specifically, the stimuli within a category shared overlapping features, leading to reduced discriminability for the decoder when distinguishing between them. Our primary objective was to assess cross-category performance rather than within-category accuracy, which may explain the observed pattern in the diagonal values.

Regarding the reference to Figure 2, we appreciate the reviewer pointing out the ambiguity. To avoid any confusion, we have removed the sentence referencing "conventions from Figure 2" in the legend for Figure 5e, as it does not contribute meaningfully to the understanding of the results.

(31) Line 473: "movement evoked response", what is meant by this?

Thank the reviewer for highlighting this point. To clarify, by "movement-evoked response," we are referring to neural activity that is driven by the animal's movements, rather than by sensory inputs. This type of response is typically stereotyped, meaning that it has a consistent, repetitive pattern associated with specific movements, such as whisking, running, or other body or facial movements.

In our study, we propose that the visually-evoked responses observed within the 150 ms time window after cue onset primarily reflect sensory inputs from the visual stimulus rather than movement-related activity. This interpretation is supported by the response timing: visual-evoked activity occurs within 100 ms of the light flash onset, a timeframe too rapid to be attributed to body or orofacial movements. Additionally, unlike stereotyped movement-evoked responses, the visual responses we observed are discriminative, varying based on specific visual features—a hallmark of sensory processing rather than motor-driven activity.

We have revised the manuscript as follows (eighth paragraph in discussion section):

“There is ongoing debate about whether cross-sensory responses in sensory cortices predominantly reflect sensory inputs or are influenced by behavioral factors, such as cue-induced body movements. A recent study shows that sound-clip evoked activity in visual cortex have a behavioral rather than sensory origin and is related to stereotyped movements(49). Several studies have demonstrated sensory neurons can encode signals associated with whisking(50), running(51), pupil dilation(52) and other movements(53). In our study, the responses to visual stimuli in the auditory cortex occurred primarily within a 100 ms window following cue onset. suggests that visual information reaches the AC through rapid pathways. Potential candidates include direct or fast cross-modal inputs, such as pulvinar-mediated pathways(8) or corticocortical connections(5，54), rather than slower associative mechanisms. This early timing suggests that the observed responses were less likely modulated by visually-evoked body or orofacial movements, which typically occur with a delay relative to sensory cue onset(55). ”

(32) Line 638-642: It is stated that a two-tailed permutation test is done. The cue selectivity can be significantly positive and negative, relative to a shuffle distribution. This is excellent. But then it is stated that if the observed ROC value exceeds the top 5% of the distribution it is deemed significant, which corresponds to a one-tailed test. How were significantly negative ROC values detected with p<0.05?

Thank you for pointing this out. We confirm that a two-tailed permutation test was indeed used to evaluate cue selectivity. In this approach, significance is determined by comparing the observed ROC value to both tails of the shuffle distribution. Specifically, if the observed ROC value exceeds the top 2.5% or falls below the bottom 2.5% of the distribution, it is considered significant at p< 0.05. This two-tailed test ensures that both significantly positive and significantly negative cue selectivity values are identified.

To clarify this in the manuscript, we have revised the text as follows:

“This generated a distribution of values from which we calculated the probability of our observed result. If the observed ROC value exceeds the top 2.5% of the distribution or falls below the bottom 2.5%, it was deemed significant (i.e., p < 0.05).”

(33) Line 472: the cited paper (reference 52) actually claims that motor-related activity in the visual cortex has an onset before 100ms and thus does not support your claim that the time window precludes any confound of behaviorally mediated activity. Furthermore, that study and reference 47 show that sensory stimuli could be discriminated based on the cue-evoked body movements and are discriminative. A stronger counterargument would be that both studies show very fast auditory-evoked body movements, but only later visually-evoked body movements.

We appreciate the reviewer’s comments. As Lohuis et al. (reference 55) demonstrated, activity in the visual cortex (V1) can reflect distinct visual, auditory, and motor-related responses, with the latter often dissociable in timing. In their findings, visually-evoked movement-related activity arises substantially later than the sensory visual response, generally beginning around 200 ms post-stimulus onset. In contrast, auditory-evoked activity in A1 occurs relatively early.

We have revised the manuscript as follows (eighth paragraph in discussion section):

“A recent study shows that sound-clip evoked activity in visual cortex have a behavioral rather than sensory origin and is related to stereotyped movements(49). ...This early timing suggests that the observed responses were less likely modulated by visually-evoked body or orofacial movements, which typically occur with a delay relative to sensory cue onset(55). ”

(34) The training order (multisensory cue first) is important to briefly mention in the main text.

We appreciate the reviewer’s suggestion and have added this information to the main text. The revised text now reads:

“The training proceeded in two stages. In the first stage, which typically lasted 3-5 weeks, rats were trained to discriminate between two audiovisual cues. In the second stage, an additional four unisensory cues were introduced, training the rats to discriminate a total of six cues.”

(35) Line 542: As I understand the multisensory rats were trained using the multisensory cue first, so different from the training procedure in the unisensory task rats where auditory trials were learned first.

Thank you for pointing this out. You are correct that, in the unisensory task, rats were first trained to discriminate auditory cues, followed by visual cues. To improve clarity and avoid any confusion, we have removed the sentence "Similar to the multisensory discrimination task" from the revised text.

(36) Line 546: Can you note on how the rats were motivated to choose both ports, or whether they did so spontaneously?

Thank you for your insightful comment. The rats' port choice was spontaneous in this task, as there was no explicit motivation required for choosing between the ports. We have clarified this point in the text to address your concern. The revised sentence now reads:

“They received a water reward at either port following the onset of the cue, and their port choice was spontaneous.”

(37) It is important to mention in the main text that the population decoding is actually pseudopopulation decoding. The interpretation is sufficiently important for interpreting the results.

Thank you for this valuable suggestion. We have revised the text to specify "pseudo-population" instead of "population" to clarify the nature of our decoding analysis. The revised text now reads:

“Our multichannel recordings enabled us to decode sensory information from a pseudo-population of AC neurons on a single-trial basis. Using cross-validated support vector machine (SVM) classifiers, we examined how this pseudo-population discriminates between stimuli.”

(38) The term modality selectivity for the description of the multisensory interaction is somewhat confusing. Modality selectivity suggests different responses to the visual or auditory trials. The authors could consider a different terminology emphasizing the multisensory interaction effect.

Thank you for your insightful comment. We have replaced " modality selectivity " with " multisensory interactive index " (MSI). This term more accurately conveys a tendency for neurons to favor multisensory stimuli over individual sensory modalities (visual or auditory alone).

(39) In Figures 3 e and g the color code is different from adjacent panels b and c and is to be deciphered from the legend. Consider changing the color coding, or highlight to the reader that the coloring in Figures 3b and c is different from the color code in panels 3 e and g.

We appreciate the reviewer’s observation. However, we believe that a change in the color coding is not necessary. Figures 3e and 3g differentiate symbols by both shape and color, ensuring accessibility and clarity. This is clearly explained in the figure legend to guide readers effectively.

(40) Figure S2b: was significance tested here?

Yes, we did it.

(41) Figure S2d: test used?

Yes, test used.

(42) Line 676: "as appropriate", was a normality test performed prior to statistical test selection?

In our analysis, we assessed normality before choosing between parametric (paired t-test) and non-parametric (Wilcoxon signed-rank test) methods. We used the Shapiro-Wilk test to evaluate the normality of the data distributions. When data met the assumption of normality, we applied the paired t-test; otherwise, we used the Wilcoxon signed-rank test.

Thank you for pointing this out. We confirm that a normality test was performed prior to the selection of the statistical test. Specifically, we used the Shapiro-Wilk test to assess whether the data distributions met the assumption of normality. Based on this assessment, we applied the paired t-test for normally distributed data and the Wilcoxon signed-rank test for non-normal data.

To ensure clarity, we update the "Statistical Analysis" section of the manuscript with the following revised text:

“For behavioral data, such as mean reaction time differences between unisensory and multisensory trials, cue selectivity and mean modality selectivity across different auditory-visual conditions, comparisons were performed using either the paired t-test or the Wilcoxon signed-rank test. The Shapiro-Wilk test was conducted to assess normality, with the paired t-test used for normally distributed data and the Wilcoxon signed-rank test for non-normal data.”

(43) Line 679: incorrect, most data is actually represented as mean +- SEM.

Thank you for pointing this out. In the Results section, we report data as mean ± SD for descriptive statistics, while in the figures, the error bars typically represent the standard error of the mean (SEM) to visually indicate variability around the mean. We have specified in each figure legend whether the error bars represent SD or SEM.

**Reviewer #2 (Recommendations for the authors):**
(1) Line 182 - here it sounds like you mean your classifier was trained to decode the modality of the stimulus, when I think what you mean is that you decoded the stimulus contingencies using A/V/AV cues?

Thank you for pointing out this potential misunderstanding. We would like to clarify that the classifier was trained to decode the stimulus identity (e.g., A^3k^ vs. A^10k^ for auditory stimuli, V^hz^ vs. V^vt^ for visual stimuli, and A^3k^V^hz^ vs. A^10k^V^vt^ for multisensory stimuli) rather than the modality of the stimulus. The goal of the analysis was to determine how well the pseudo-population of AC neurons could distinguish between individual stimuli within the same modality. We have revised the relevant text in the revised manuscript to ensure this distinction is clear. Please see the following:

“Our multichannel recordings enabled us to decode sensory information from a pseudo-population of AC neurons on a single-trial basis. Using cross-validated support vector machine (SVM) classifiers, we examined how this pseudo-population discriminates stimulus identity (e.g., A^3k^ vs. A^10k^ for auditory stimuli, V^hz^ vs. V^vt^ for visual stimuli, A^3k^V^hz^ vs. A^10k^V^vt^ for multisensory stimuli).”

(2) Lines 256 - here the authors look to see whether incorrect trials diminish audiovisual integration. I would probably seek to turn the causal direction around and ask are AV neurons critical for behaviour - nevertheless, since this is only correlational the causal direction cannot be unpicked. However, the finding that contralateral responses per se do not result in enhancement is a key control. Showing that multisensory enhancement is less on error trials is a good first step to linking neural activity and perception, but I wonder if the authors could take this further however by seeking to decode choice probabilities as well as stimulus features in an attempt to get a little closer to addressing the question of whether the animals are using these responses for behaviour.

Thank you for your comment and for highlighting the importance of understanding whether audiovisual (AV) neurons are critical for behavior. As you noted, the causal relationship between AV neural activity and behavioral outcomes cannot be directly determined in our current study due to its correlational nature. We agree that this is an important topic for future exploration. In our study, we examined how incorrect trials influence multisensory enhancement. Our findings show that multisensory enhancement is less pronounced during error trials, providing an initial link between neural activity and behavioral performance. To address your suggestion, we conducted an additional analysis comparing auditory and multisensory selectivity between correct and incorrect choice trials. As shown in Supplementary Fig. 7, both auditory and multisensory selectivity were significantly lower during incorrect trials. This result highlights the potential role of these neural responses in decision-making, suggesting they may extend beyond sensory processing to influence choice selection. We have cited this figure in the Results section as follows: (the paragraph regarding Impact of incorrect choices on audiovisual integration):

“Overall, these findings suggest that the multisensory perception reflected by behavioral choices (correct vs. incorrect) might be shaped by the underlying integration strength. Furthermore, our analysis revealed that incorrect choices were associated with a decline in cue selectivity, as shown in Supplementary Fig. 7.”

We acknowledge your suggestion to decode choice probabilities alongside stimulus features as a more direct approach to exploring whether animals actively use these neural responses for behavior. Unfortunately, in the current study, the low number of incorrect trials limited our ability to perform such analyses reliably. Nonetheless, we are committed to pursuing this direction in subsequent work. We plan to use techniques such as optogenetics in future studies to causally test the role of AV neurons in driving behavior.

(3) Figure 5E - the purple and red are indistinguishable - could you make one a solid line and keep one dashed?

We thank the reviewer for pointing out that the purple and red lines in Figure 5E were difficult to distinguish. To address this concern, we modified the figure by making two lines solid and changing the color of one square, as suggested. These adjustments enhance visual clarity and improve the distinction between them.

(4) The unisensory control training is a really nice addition. I'm interested to know whether behaviourally these animals experienced an advantage for audiovisual stimuli in the testing phase? This is important information to include as if they don't it is one step closer to linking audiovisual responses in AC to improved behavioural performance (and if they do, we must be suitably cautious in interpretation!).

Thank you for raising this important point. To address this, we have plotted the behavioral results for each animal (see Author response image 2). The data indicate that performance with multisensory cues is slightly better than with the corresponding unisensory cues. However, given the small sample size (n=3) and the considerable variation in behavioral performance across individuals, we remain cautious about drawing definitive conclusions on this matter. We recognize the need for further investigation to establish a robust link between audiovisual responses in the auditory cortex and improved behavioral performance. In future studies, we plan to include a larger number of animals and more thoroughly explore this relationship to provide a comprehensive understanding.

**Author response image 2. sa4fig2:** 

(5) Line 339 - I don't think you can say this leads to binding with your current behaviour or neural responses. I would agree there is a memory trace established and a preferential linking in AC neurons.

We thank the reviewer for raising this important point. In the revised manuscript, we have clarified that our data suggest the formation of a memory trace and preferential linking in AC neurons. The text has been updated to emphasize this distinction. Please see the revised section below (first paragraph in Discussion section).

“Interestingly, a subset of auditory neurons not only developed visual responses but also exhibited congruence between auditory and visual selectivity. These findings suggest that multisensory perceptual training establishes a memory trace of the trained audiovisual experiences within the AC and enhances the preferential linking of auditory and visual inputs. Sensory cortices, like AC, may act as a vital bridge for communicating sensory information across different modalities.”